# CDTP: A Large-Scale Chinese Data-Text Pair Dataset for Comprehensive Evaluation of Chinese LLMs

## Abstract

Large Language Models (LLMs) have achieved remarkable success across a wide range of natural language processing tasks. However, Chinese LLMs face unique challenges, primarily due to the dominance of unstructured free text and the lack of structured representations in Chinese corpora. While existing benchmarks for LLMs partially assess Chinese LLMs, they are still predominantly English-centric and fail to address the unique linguistic characteristics of Chinese, lacking structured datasets essential for robust evaluation. To address these challenges, we present a **C**omprehensive **B**enchmark for **E**valuating **C**hinese **L**arge **L**anguage **M**odels (CB-ECLLM) based on the newly constructed Chinese Data-Text Pair (CDTP) dataset. Specifically, CDTP comprises over 7 million aligned text pairs, each consisting of unstructured text coupled with one or more corresponding triples, alongside a total of 15 million triples spanning four critical domains. The core contributions of CDTP are threefold: (i) enriching Chinese corpora with high-quality structured information; (ii) enabling fine-grained evaluation tailored to knowledge-driven tasks; and (iii) supporting multi-task fine-tuning to assess generalization and robustness across scenarios, including Knowledge Graph Completion, Triple-to-Text generation, and Question Answering. Furthermore, we conduct rigorous evaluations through extensive experiments and ablation studies to assess the effectiveness, Supervised Fine-Tuning (SFT), and robustness of the benchmark. To support reproducible research, we offer an open-source codebase and outline potential directions for future investigations based on our insights.

## 1 Introduction

Large Language Models (LLMs) (Pan et al., 2024; Chang et al., 2024) have demonstrated exceptional performance across a broad range of Natural Language Processing (NLP) tasks (Wang et al., 2024a; Yan et al., 2025; 2024), positioning themselves as pivotal milestones on the path toward Artificial General Intelligence (AGI) (Pei et al., 2019; Fei et al., 2022). However, achieving AGI requires mastering not only proficiency in widely studied languages like English, but also mastery of complex linguistic systems such as Chinese, which present unique and substantial challenges.

Chinese poses distinctive challenges for LLMs, largely because its corpora are dominated by unstructured free text and lack explicit structured annotations such as entity–attribute–value triples, which are essential for learning reliable entity–relation mappings (Liu et al., 2022; Zhu et al., 2022). Table 1 illustrates this issue: a single Chinese expression can yield multiple plausible interpretations depending on context. This absence of structured signals constrains performance on a wide range of knowledge-intensive tasks, including knowledge retrieval (Li et al., 2025), relation inference (Wang et al., 2024b), and text generation (Li et al., 2024a). In these settings, models are required to extract or memorize facts from lengthy unstructured text, a process that increases

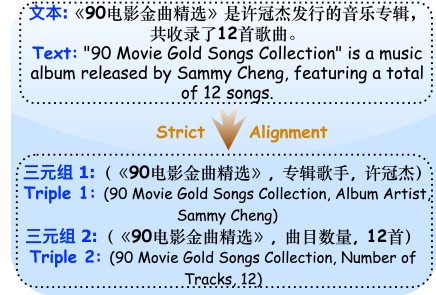

Figure 1: The example of our proposed Chinese Data-Text Pair (CDTP) dataset.

Table 1: Examples illustrating polysemy and structural ambiguity in Chinese text. Each case shows one surface form with two possible interpretations.

| Chinese Expression | Interpretation 1 | Interpretation 2 |
|---|---|---|
| 行 | 行动 | 行业 |
| | to do/go | profession/industry |
| 武汉市长江大桥 | 武汉市长/江大桥 | 武汉市/长江大桥 |
| | the mayor of Wuhan named Jiang Daqiao | the Yangtze River Bridge in Wuhan |

training difficulty, propagates errors, weakens generalization across domains, and ultimately reduces interpretability.

To bridge this gap, we construct the Chinese Data-Text Pair (CDTP) dataset, which tightly aligns structured triples with corresponding natural language text. CDTP is developed through rigorous collection and cleaning procedures to guarantee strict alignment, high-quality content, and broad domain coverage. In total, it comprises over 7 million text samples and their corresponding 15 million triples, spanning four major domains: `History and Politics`, `Humanities and Society`, `Technology and Economics`, and `Nature and Environment`. This large-scale, systematically curated resource provides a solid foundation for evaluating the interplay between structured knowledge and unstructured text, and its overall format is illustrated in Figure 1.

Moreover, existing benchmarks for evaluating LLMs, such as CMMLU (Li et al., 2023), C-Eval (Huang et al., 2023), and SuperCLUE (Xu et al., 2023), are largely shaped by English-centric task design and fail to capture challenges unique to Chinese, including the scarcity of structured annotations and the prevalence of ambiguous expressions. Consequently, they provide only an incomplete assessment of Chinese LLMs in knowledge-intensive tasks requiring relational reasoning and factual accuracy. To overcome these limitations, we further present the **C**omprehensive **B**enchmark for **E**valuating **C**hinese **L**arge **L**anguage **M**odels (CB-ECLLM), constructed on top of CDTP to provide systematic and knowledge-driven evaluation. CB-ECLLM targets three representative tasks, including `Knowledge Graph Completion(KGC)` (Wang et al., 2024a; 2023a), `Triple-to-Text Generation(T2T)` (Li et al., 2021), and `Question Answer(QA)` (He et al., 2024). Together, these tasks capture the interplay between structured knowledge and natural language, providing a more targeted evaluation of Chinese LLMs.

Building on these tasks, we conducted experiments to evaluate Chinese LLMs across three key dimensions: effectiveness, Supervised Fine-Tuning (SFT) (Dong et al., 2024; Lu et al., 2024), and robustness. For **effectiveness**, we evaluate the performance of 8 Chinese LLMs across 3 tasks on 4 distinct sub-datasets. This evaluation assesses how well the Chinese LLMs generate accurate responses by integrating structured knowledge, and how they perform across different domains and task-specific challenges. Regarding **SFT**, we conduct further experiments to evaluate the impact of incorporating additional labeled data on the performance of Chinese LLMs. In terms of **robustness**, we assess the stability of the SFT Chinese LLMs by testing their performance on Out-Of-Distribution (OOD) data. This aims to explore how well LLMs can maintain accuracy and reliability when faced with data that deviates from the training distribution. To summarize, the primary contributions of this paper is three-fold:

- **Comprehensive Benchmark.** We introduce a comprehensive benchmark tailored to the unique linguistic and cultural complexities of Chinese, such as context-dependent disambiguation, morphological segmentation, and idiomatic expression comprehension, addressing a critical gap in existing evaluations;

- **High-Quality Aligned Dataset.** To the best of our knowledge, we have constructed the largest Chinese Data-to-Text Pair (CDTP) dataset, comprising 7 million meticulously aligned text pairs, each consisting of unstructured text paired with one or more corresponding triples, totaling 15 million triples across four major domains;

- **Multi-Dimensional Evaluation.** We systematically evaluate Chinese LLMs from multiple perspectives, including performance on diverse tasks and datasets, generalizability across domains and tasks, and robustness against Out-Of-Distribution (OOD) data.

## 2 RELATED WORK

Comprehensive evaluation of LLMs typically encompasses three core dimensions: linguistic competence, factual accuracy, and domain-specific knowledge. Existing benchmarks can be broadly categorized based on their target language into Chinese-oriented and English-oriented benchmarks.

**Chinese-oriented Benchmarks.** Benchmarks such as CMMLU (Li et al., 2023) and C-Eval (Huang et al., 2023) provide large-scale multiple-choice evaluations covering various disciplines. Super-CLUE (Xu et al., 2023) extends this by including agent-centric and safety-sensitive tasks. More recently, GraphEval (Liu et al., 2024) proposes to evaluate the factuality of LLMs by comparing model outputs against a large-scale Chinese knowledge graph, avoiding expensive human annotations. Similarly, KG-SFT (Zhao et al., 2024) demonstrates that fine-tuning LLMs with structured knowledge graph explanations enhances their performance in QA tasks, especially in low-resource domains. Beyond evaluation, several efforts have explored knowledge graph construction in Chinese. For instance, TechGPT-2.0 (Wang et al., 2025) focuses on entity and relation extraction using LLMs in Chinese, targeting complex domains such as law and medicine. Another line of work explores the effect of fine-tuning vs. prompting strategies for transforming unstructured text into structured triples (Chen et al., 2024). Further details on comparison with existing Chinese datasets can be found in Appendix B.1.

However, none of these works offer a large-scale, domain-diverse, and strictly aligned triple-text resource that can serve both as a benchmark and as supervised data for structured generation and reasoning in Chinese.

**English-oriented Benchmarks.** In English, datasets such as WebNLG (Gardent et al., 2017), KELM (Agarwal et al., 2021), and KGTEXT (Annasamy et al., 2023) provide aligned triple-text pairs for text generation, supporting evaluation of LLMs' capabilities in knowledge-grounded generation. Recent work, such as KG-LLM (Yao et al., 2024), proposes treating knowledge graph triples as sequences and uses LLMs for tasks like triple classification and relation prediction, demonstrating strong results even with mid-sized models. A recent survey by Khorashadizadeh et al. (2024) categorizes the evolving interplay between LLMs and KGs, covering tasks like KG question answering, validation, and descriptive text generation.

While these English-centric benchmarks and models offer valuable insights, they do not account for the linguistic and cultural challenges unique to Chinese, such as character-level ambiguity, idiomatic expressions, and implicit subjects. Moreover, no existing English benchmark provides Chinese-equivalent scale and alignment rigor tailored to LLM evaluation.

## 3 CHINESE DATA-TO-TEXT PAIR (CDTP) DATASET

The Chinese Data-to-Text Pair (CDTP) dataset is constructed through a systematic methodology consisting of two sequential phases: data collection and data processing. In the data collection phase, we compile a raw data corpus of approximately 27 million entries from various online encyclopedic sources, such as *Baidu Baike* and *Sogou Baike*. In the data processing phase, we refine this data through data cleaning and quality improvement to ensure consistency and factual accuracy. Furthermore, we adopt targeted strategies to increase data diversity and ensure balanced coverage across multiple domains. This section provides a detailed description of the CDTP dataset construction pipeline, including statistical analysis and category-level organization.

### 3.1 DATA COLLECTION

The data collection pipeline is designed to construct a large-scale KG-Text pair dataset from heterogeneous sources, ensuring broad coverage and high-quality alignment. The pipeline consists of two main stages: seed entry generation, followed by data crawling and expansion. This process yields approximately 30 million triples involving around 15 million unique entities.

**Seed Entry Generation.** The data collection process begins with aggregating a set of seed entries from three sources: OwnThink (Li et al., 2024b), KgClue (Wang et al., 2023b), and WudaoCorpora (Yuan et al., 2021). This step yields a total of 56 million seed entries, which serve as the foundation for subsequent data crawling. The diversity of these sources ensures broad coverage across various domains and contexts.

**Data Crawling and Expansion.** Baike websites typically offer concise textual descriptions and structured relation tables, such as infoboxes containing attribute-value pairs, for individual entries. These tables provide a rich source of high-quality relational information that can be directly converted into triples, making them particularly valuable for structured knowledge extraction. Leveraging these features, we develop a customized Baike crawler that systematically retrieves both textual descriptions and triples for each seed entry from a precompiled list. To ensure comprehensive coverage and semantic diversity, the crawler employs a depth-first search strategy: it initiates from the Baike page of a seed entry and recursively traverses hyperlinked entities, capturing a wide array of related concepts and relations. This recursive expansion enables the system to uncover latent semantic connections across entries, effectively transforming a relatively small seed set into a large-scale, rich, and diverse KG with extensive entity and relation coverage.

## 3.2 DATA PROCESSING

The data processing pipeline refines raw collections to ensure accuracy, consistency, and relevance, comprising two main stages: data cleaning and quality enhancement. In the cleaning stage, we remove noise and inconsistencies using string matching, position-based filtering, and rule-based deduplication, while the enhancement stage improves reliability through statistical filtering, expert validation, and external knowledge verification, ensuring precise alignment between structured triples and textual descriptions with high relational accuracy.

**Data Cleaning.** Given the encyclopedic nature of Baike-sourced text data, we specifically target two prevalent types of noise in our cleaning process: (1) *redundant triples*—relation assertions unsupported by textual evidence, and (2) *redundant descriptions*—repetitive or irrelevant content that deviates from the corresponding structured knowledge, thus ensuring better alignment between text and triples.

*Redundant Triples.* Redundant triples refer to triples that are not mentioned in the accompanying text description. As illustrated in Table 2, the triple "(90电影金曲精选, 专辑语言, 粤语) (Golden Songs in 90s Movies, album language, Cantonese)" conveys factual information about the album but is not explicitly reflected in the corresponding textual description, and is therefore treated as a redundant triple. To ensure alignment, we examine each data pair by checking whether the semantic content of each triple is supported by the accompanying text. Triples that lack textual grounding are removed. If all triples in a pair are filtered out, the entire data pair is discarded. This filtering procedure ensures strong semantic alignment between structured knowledge and natural language, laying a reliable foundation for downstream tasks. Additionally, discarded triples may serve as auxiliary data for text generation or data augmentation.

Table 2: Illustrative examples of redundant triples (highlighted in red).

| | |
|---|---|
| **Triple 1** | (90电影金曲精选, 专辑语言，粤语)
(Golden Songs in 90s Movies, album_language, Cantonese) |
| **Triple 2** | (90电影金曲精选, 专辑歌手, 许冠杰)
(Golden Songs in 90s Movies, allbum_singer, Guanjie Xu) |
| **Triple 3** | (90电影金曲精选, 曲目数量, 12首)
(Golden Songs in 90s Movies, number_of_songs, 12) |
| **Text** | 《90电影金曲精选》是许冠杰发行的音乐专辑，共收录了12首歌曲。
Golden Songs in 90s Movies is a music album released by Guanjie Xu, which contains 12 songs. |

*Redundant Descriptions.* Redundant descriptions contain information present in the text but absent from the relation triples. As shown in Table 3, text descriptions often contain additional factual details, such as "2.4 inches" or "3G mobile phone", that are not represented in the associated triples. To identify and eliminate such redundancy, we compute a position index by locating the character offset of the last matched triple and normalizing it by the total length of the text. Pairs with an index above 0.75, indicating that triples appear predominantly towards the end—are retained; otherwise, they are discarded. For the retained pairs, we inspect the remaining text beyond the last triple to detect unmatched relation mentions by cross-referencing with all Baike-extracted relations. Additionally, in enumerated structures separated by punctuation, we require that all items are grounded in the relation triples. If any listed item lacks a corresponding triple, the data pair is excluded to maintain strict alignment.

Table 3: Examples of redundant descriptions (marked in red).

| | |
|---|---|
| **Triple 1** | (黑莓 9220，操作系统，Blackberry OS 7)
(BlackBerry 9220, operating_system, Blackberry OS 7) |
| **Triple 2** | (黑莓 9220，上市日期，2012年)
(BlackBerry 9220, launch_date, 2012) |
| **Text** | 黑莓 9220是2012年上市的一款支持Blackberry OS 7操作系统的2.4英寸的3G手机。
BlackBerry 9220 is a 2.4-inch 3G mobile phone that supports BlackBerry Os 7 operating system, which was launched in 2012. |

**Data Quality Improvement.** To improve data quality and address the long-tail distribution in KGs, we employ three complementary strategies: *Relation Filtering.* We retain only high-frequency relation types to reduce noise and emphasize semantically meaningful connections (Appendix B.2). *Manual Verification.* Seven trained annotators independently review each relation type, with double annotation ensuring over 90% agreement and filtering out low-quality triples. *Search Validation.* For each candidate triple, we retrieve the top 10 external web results and retain it only if both the head and tail entities co-occur, ensuring contextual accuracy. The final data is categorized into four domains—*History and Politics*, *Humanities and Society*, *Technology and Economics*, and *Nature and Environment*—to support domain-specific evaluation. These measures collectively improve dataset quality and establish a reliable benchmark for evaluating Chinese LLMs. Further details on data statistics can be found in Appendix B.2 and B.3.

## 4 BENCHMARK DESIGN

In this section, we provide a comprehensive overview of CB-ECLLM, covering benchmark settings (Sec. 4.1), and evaluation metrics & implementation details (Sec. 4.2).

### 4.1 BENCHMARK SETTINGS

**Benchmark Datasets.** To evaluate model performance across diverse domains, we construct our benchmark using four representative sub-datasets drawn from the full CDTP collection (7M pairs): *CDTP_HP (History and Politics)*, *CDTP_HS (Humanities and Society)*, *CDTP_TE (Technology and Economics)*, and *CDTP_NE (Nature and Environment)*. For the experiments in this paper, each sub-dataset is split into 80% training and 20% evaluation. To ensure tractable and reproducible experiments while preserving data diversity, we report results on a stratified 10K subset drawn from the full CDTP dataset. This subset preserves key distributions (domain, entity type, predicate frequency, and text length), is sampled with a fixed pseudorandom seed, and follows the same 80%/20% split per domain.

**Evaluation Models.** To evaluate the performance of Chinese LLMs on our proposed CDTP dataset, we selected eight state-of-the-art models with diverse architectures and training paradigms, including both general-purpose and Chinese-focused LLMs. The selected LLMs are GLM-4-9B (GLM et al., 2024), Yi-9B (Young et al., 2024), Qwen1.5-7B (Bai et al., 2023), InternLM2-7B (Cai et al., 2024), Llama-3-8B (Grattafiori et al., 2024), Baichuan2-7B (Yang et al., 2023), Phi-2 (Javaheripi et al., 2023), and DeepSeek-7B (Bi et al., 2024). Details of these Chinese LLMs are provided in Appendix C.1.

**Benchmark Tasks.** To fully unlock the potential of the CDTP dataset, we conduct a comprehensive evaluation of Chinese LLMs based on three tasks, including Question Answer (QA), Knowledge Graph Completion (KGC), and Triple to Text Generation (T2T). For QA and KGC, each query is cast as a multiple-choice problem with one correct answer and nine distractors. Distractors are drawn from KG neighbors and type-/frequency-matched entities, filtered for semantic feasibility, and enriched with polysemy-targeted candidates to capture Chinese-specific ambiguities. This design enables these tasks to assess both relational reasoning and robustness under contextual ambiguity. Detailed task definitions and construction procedures are provided in Appendix C.2.

### 4.2 EVALUATION METRICS AND IMPLEMENTATION DETAILS

**Evaluation Metrics.** Several commonly used metrics are employed to evaluate the performance of Chinese LLMs across the three tasks: QA, KGC, and T2T. For QA and KGC, we use Mean Reciprocal Rank (MRR), Hits@1, and F1 Score. For the T2T task, BLEU, ROUGE, and METEOR are applied. Detailed descriptions and calculation methods for these metrics are provided in Appendix C.3.

**Supervised Fine-Tuning (SFT) Parameters Settings.** To validate the quality of our proposed CDTP dataset, SFT experiments were conducted on 8 LLMs using DeepSpeed (Rasley et al., 2020). Each dataset category was fine-tuned independently with the following configurations: {Training Framework: DeepSpeed with ZeRO Stage 2; Batch Size: 8; Learning Rate: $9.65 \times 10^{-6}$ with a cosine learning rate scheduler; Max Sequence Length: 2048; Mixed Precision: BF16; Number of Epochs: 3}. All models are fine-tuned on the SFT datasets using 8 NVIDIA H100 GPUs.

Table 4: Evaluation Results of Eight Chinese LLMs on Four Datasets for Three Tasks. The best three results are highlighted by  1st ,  2nd , and  3rd .

| Dataset | Task | Metrics | GLM-4-9B | Yi-9B | Qwen1.5-7B | InternLM2-7B | Llama-3-8B | Baichuan2-7B | DeepSeek-7B | Phi-2 |
|---|---|---|---|---|---|---|---|---|---|---|
| CDTP_HP | KGC | MRR | 0.4417 | 0.6659 | 0.5458 | 0.4906 | 0.4538 | 0.4607 | 0.5364 | 0.3305 |
| | | Hits@1 | 0.3445 | 0.5250 | 0.4005 | 0.3210 | 0.2720 | 0.2385 | 0.3710 | 0.1911 |
| | | F1 Score | 0.5125 | 0.6885 | 0.5719 | 0.4860 | 0.4277 | 0.3851 | 0.5412 | 0.3209 |
| | QA | MRR | 0.2594 | 0.4818 | 0.3469 | 0.3712 | 0.2911 | 0.3236 | 0.2942 | 0.3078 |
| | | ACC | 0.1870 | 0.3695 | 0.2425 | 0.2475 | 0.1835 | 0.1695 | 0.2005 | 0.1970 |
| | | F1 Score | 0.3151 | 0.5396 | 0.3903 | 0.3968 | 0.3101 | 0.2899 | 0.3340 | 0.3292 |
| | T2T | BLEU | 0.2405 | 0.2647 | 0.1471 | 0.2211 | 0.2058 | 0.1542 | 0.2502 | 0.1089 |
| | | ROUGE_1 | 0.7073 | 0.7253 | 0.5516 | 0.6968 | 0.6784 | 0.5569 | 0.6767 | 0.5680 |
| | | ROUGE_L | 0.6448 | 0.6655 | 0.4969 | 0.6223 | 0.6099 | 0.5100 | 0.6246 | 0.5235 |
| | | METEOR | 0.6972 | 0.7030 | 0.4029 | 0.6608 | 0.6462 | 0.4872 | 0.6442 | 0.4556 |
| CDTP_HS | KGC | MRR | 0.4927 | 0.6805 | 0.5363 | 0.4613 | 0.4449 | 0.4457 | 0.5610 | 0.2959 |
| | | Hits@1 | 0.3937 | 0.5430 | 0.3975 | 0.2925 | 0.2640 | 0.2070 | 0.4140 | 0.1621 |
| | | F1 Score | 0.5650 | 0.7038 | 0.5689 | 0.4526 | 0.4177 | 0.3430 | 0.5856 | 0.2790 |
| | QA | MRR | 0.2319 | 0.4535 | 0.3072 | 0.3555 | 0.2915 | 0.3049 | 0.3012 | 0.2932 |
| | | ACC | 0.1600 | 0.3310 | 0.1960 | 0.2210 | 0.1675 | 0.1352 | 0.1990 | 0.1715 |
| | | F1 Score | 0.2759 | 0.4974 | 0.3278 | 0.3620 | 0.2869 | 0.2382 | 0.3319 | 0.2928 |
| | T2T | BLEU | 0.2235 | 0.2556 | 0.1361 | 0.2228 | 0.1964 | 0.1481 | 0.2230 | 0.1012 |
| | | ROUGE_1 | 0.6992 | 0.7174 | 0.5527 | 0.7009 | 0.6694 | 0.5533 | 0.6628 | 0.5706 |
| | | ROUGE_L | 0.6408 | 0.6586 | 0.5025 | 0.6307 | 0.6079 | 0.5054 | 0.6128 | 0.5181 |
| | | METEOR | 0.6910 | 0.7086 | 0.4052 | 0.6696 | 0.6411 | 0.4771 | 0.6201 | 0.4614 |
| CDTP_NE | KGC | MRR | 0.4693 | 0.7249 | 0.6002 | 0.5105 | 0.4527 | 0.4602 | 0.6413 | 0.2805 |
| | | Hits@1 | 0.3900 | 0.6055 | 0.4735 | 0.3460 | 0.2690 | 0.1940 | 0.5145 | 0.1490 |
| | | F1 Score | 0.5612 | 0.7543 | 0.6427 | 0.5141 | 0.4240 | 0.3249 | 0.6794 | 0.2594 |
| | QA | MRR | 0.3261 | 0.5106 | 0.3149 | 0.4215 | 0.3385 | 0.4023 | 0.3532 | 0.3028 |
| | | ACC | 0.2650 | 0.4235 | 0.2320 | 0.3145 | 0.2245 | 0.2805 | 0.2750 | 0.1925 |
| | | F1 Score | 0.4190 | 0.5950 | 0.3766 | 0.4785 | 0.3667 | 0.4381 | 0.4314 | 0.3229 |
| | T2T | BLEU | 0.2146 | 0.2213 | 0.1213 | 0.1786 | 0.2079 | 0.1209 | 0.2153 | 0.0851 |
| | | ROUGE_1 | 0.6910 | 0.6912 | 0.5399 | 0.6667 | 0.6789 | 0.5412 | 0.6508 | 0.5502 |
| | | ROUGE_L | 0.6535 | 0.6505 | 0.5029 | 0.6147 | 0.6367 | 0.5076 | 0.6210 | 0.5108 |
| | | METEOR | 0.6791 | 0.6705 | 0.4008 | 0.6225 | 0.6502 | 0.4693 | 0.6136 | 0.4259 |
| CDTP_TE | KGC | MRR | 0.4856 | 0.6705 | 0.4973 | 0.4619 | 0.4424 | 0.4488 | 0.5305 | 0.3112 |
| | | Hits@1 | 0.3910 | 0.5375 | 0.3620 | 0.2840 | 0.2610 | 0.2240 | 0.3675 | 0.1856 |
| | | F1 Score | 0.5622 | 0.6992 | 0.5316 | 0.4424 | 0.4140 | 0.3660 | 0.5375 | 0.3130 |
| | QA | MRR | 0.3070 | 0.5076 | 0.3486 | 0.4045 | 0.3147 | 0.3620 | 0.3514 | 0.3130 |
| | | ACC | 0.2199 | 0.4060 | 0.2425 | 0.2775 | 0.1910 | 0.2075 | 0.2585 | 0.1900 |
| | | F1 Score | 0.3605 | 0.5775 | 0.3903 | 0.4344 | 0.3207 | 0.3437 | 0.4108 | 0.3193 |
| | T2T | BLEU | 0.2600 | 0.2740 | 0.1665 | 0.2395 | 0.2446 | 0.1550 | 0.2552 | 0.1321 |
| | | ROUGE_1 | 0.7169 | 0.7295 | 0.5875 | 0.7077 | 0.7031 | 0.5409 | 0.6886 | 0.6032 |
| | | ROUGE_L | 0.6623 | 0.6715 | 0.5364 | 0.6415 | 0.6440 | 0.5003 | 0.6406 | 0.5584 |
| | | METEOR | 0.7165 | 0.7244 | 0.4608 | 0.6795 | 0.6854 | 0.4643 | 0.6638 | 0.4949 |

# 5 EXPERIMENTAL RESULTS

In this section, we introduce the experimental setup and discuss the experimental results in CU-ECLLM benchmark. Specifically, we aim to answer the following research questions: ● **RQ1 (Effectiveness)**: How do different Chinese LLMs perform under various datasets for three tasks? (Sec. 5.1) ● **RQ2 (SFT)**: How effective are different Chinese LLMs in fine-tuning Chinese LLMs on these labeled data improves their ability to handle the tasks? (Sec. 5.2) ● **RQ3 (Robustness)**: How well LLMs can maintain accuracy when faced with out-of-distribution (OOD) data? (Sec. 5.3),Appendix D.1 is the experiment of LLM with different parameters.

## 5.1 RQ1: PERFORMANCE COMPARISON

**Experiment Design.** We comprehensively evaluate and compare the performance of eight Chinese LLMs across three tasks—Knowledge Graph Completion (KGC), Question Answer (QA), and Triple to Text Generation (T2T)—using four distinct datasets.

**Experimental Results.** Table 4 and Table 10 show the performance comparison in terms of three tasks on four datasets, and Figure 2 provides a line graph to compare the performance between QA and KGC. Moreover, the bar graph illustrating the performance rankings of Chinese LLMs with respect to their parameter sizes is shown in Figure 3. We have the following observations.

**Observation ❶**: **Performance varies significantly across tasks and datasets.** As shown in Table 4, some Chinese LLMs excel in T2T tasks but underperform in KGC or QA tasks. For instance, InternLM2-7B performs well in T2T tasks, achieving METEOR scores above 0.66 on CDTP_TE and CDTP_HS, but its performance in QA tasks, such as Hits@1 and MRR, is less impressive. Furthermore, the performance varies across different datasets. For example, in the KGC task, some

Table 5: Evaluation Results of Eight Chinese LLMs on Four Datasets for Three Tasks After Supervised Fine-Tuning (SFT). The best three results are highlighted by 1st , 2nd , and 3rd .

| Datasets | Tasks | Metrics | Supervised Fine-Tuning (SFT) | | | | | | | |
|---|---|---|---|---|---|---|---|---|---|---|
| | | | GLM-4-9B | Yi-9B | Qwen1.5-7B | InternLM2-7B | Llama-3-8B | Baichuan2-7B | DeepSeek-7B | Phi-2 |
| CDTP_HP | KGC | MRR | 0.6105 | 0.7965 | 0.7877 | 0.7877 | 0.7553 | 0.7823 | 0.7927 | 0.6742 |
| | | Hits@1 | 0.4516 | 0.7170 | 0.7044 | 0.7029 | 0.6595 | 0.6853 | 0.7100 | 0.5420 |
| | | F1 Score | 0.6222 | 0.8352 | 0.8266 | 0.8255 | 0.7948 | 0.8133 | 0.8304 | 0.7030 |
| | QA | MRR | 0.6331 | 0.6295 | 0.6083 | 0.6368 | 0.5954 | 0.6084 | 0.6354 | 0.4832 |
| | | ACC | 0.5625 | 0.5810 | 0.5525 | 0.5865 | 0.5370 | 0.5415 | 0.5860 | 0.3622 |
| | | F1 Score | 0.7200 | 0.7350 | 0.7118 | 0.7394 | 0.6988 | 0.7026 | 0.7390 | 0.5318 |
| | T2T | BLEU | 0.4604 | 0.4720 | 0.4672 | 0.4531 | 0.4608 | 0.4523 | 0.4622 | 0.3665 |
| | | ROUGE_1 | 0.7848 | 0.7891 | 0.7859 | 0.7778 | 0.7864 | 0.7733 | 0.7848 | 0.7542 |
| | | ROUGE_L | 0.7549 | 0.7601 | 0.7551 | 0.7467 | 0.756 | 0.7436 | 0.7544 | 0.7148 |
| | | METEOR | 0.7722 | 0.7688 | 0.7647 | 0.7551 | 0.7738 | 0.7480 | 0.7650 | 0.7456 |
| CDTP_HS | KGC | MRR | 0.5268 | 0.8051 | 0.7966 | 0.7566 | 0.7683 | 0.7845 | 0.8043 | 0.6471 |
| | | Hits@1 | 0.3537 | 0.7265 | 0.7140 | 0.6783 | 0.6760 | 0.6875 | 0.7230 | 0.5058 |
| | | F1 Score | 0.5225 | 0.8416 | 0.8331 | 0.8067 | 0.8067 | 0.8148 | 0.8392 | 0.6718 |
| | QA | MRR | 0.6184 | 0.6457 | 0.6437 | 0.6479 | 0.6051 | 0.6381 | 0.6516 | 0.5094 |
| | | ACC | 0.5470 | 0.5900 | 0.5858 | 0.5890 | 0.5388 | 0.5730 | 0.5940 | 0.4032 |
| | | F1 Score | 0.7072 | 0.7421 | 0.7388 | 0.7413 | 0.7003 | 0.7285 | 0.7453 | 0.5747 |
| | T2T | BLEU | 0.3610 | 0.3822 | 0.3735 | 0.3608 | 0.3192 | 0.3667 | 0.3740 | 0.2329 |
| | | ROUGE_1 | 0.7479 | 0.7555 | 0.7513 | 0.7440 | 0.7059 | 0.7405 | 0.7501 | 0.6967 |
| | | ROUGE_L | 0.7055 | 0.7146 | 0.7109 | 0.7023 | 0.6640 | 0.6982 | 0.7120 | 0.6423 |
| | | METEOR | 0.7279 | 0.7329 | 0.7239 | 0.7107 | 0.6840 | 0.7035 | 0.7199 | 0.6631 |
| CDTP_NE | KGC | MRR | 0.6168 | 0.8246 | 0.8245 | 0.7997 | 0.7947 | 0.8131 | 0.8238 | 0.6848 |
| | | Hits@1 | 0.4705 | 0.7615 | 0.759 | 0.7285 | 0.7147 | 0.7355 | 0.7585 | 0.5712 |
| | | F1 Score | 0.6399 | 0.8646 | 0.863 | 0.8429 | 0.8336 | 0.8476 | 0.8627 | 0.7271 |
| | QA | MRR | 0.6331 | 0.6485 | 0.6284 | 0.6418 | 0.6016 | 0.6066 | 0.6431 | 0.5393 |
| | | ACC | 0.5904 | 0.6140 | 0.5853 | 0.6055 | 0.5521 | 0.5515 | 0.6035 | 0.4697 |
| | | F1 Score | 0.7424 | 0.7608 | 0.7384 | 0.7543 | 0.7114 | 0.7109 | 0.7527 | 0.6392 |
| | T2T | BLEU | 0.3755 | 0.4041 | 0.4033 | 0.3806 | 0.3978 | 0.3899 | 0.3984 | 0.2672 |
| | | ROUGE_1 | 0.7223 | 0.7516 | 0.7481 | 0.7348 | 0.7498 | 0.7385 | 0.7501 | 0.7070 |
| | | ROUGE_L | 0.6959 | 0.7291 | 0.7249 | 0.7105 | 0.7271 | 0.7144 | 0.7278 | 0.6783 |
| | | METEOR | 0.7056 | 0.7246 | 0.7226 | 0.6967 | 0.7275 | 0.7055 | 0.7246 | 0.6800 |
| CDTP_TE | KGC | MRR | 0.5719 | 0.7929 | 0.7856 | 0.7498 | 0.7600 | 0.7752 | 0.7854 | 0.6652 |
| | | Hits@1 | 0.4047 | 0.7135 | 0.6965 | 0.6725 | 0.6680 | 0.6800 | 0.7020 | 0.5277 |
| | | F1 Score | 0.5762 | 0.8328 | 0.8211 | 0.8042 | 0.8010 | 0.8095 | 0.8249 | 0.6909 |
| | QA | MRR | 0.4363 | 0.6539 | 0.6563 | 0.6206 | 0.6248 | 0.6438 | 0.6517 | 0.5365 |
| | | ACC | 0.3273 | 0.6051 | 0.6048 | 0.5695 | 0.5663 | 0.5820 | 0.5990 | 0.4445 |
| | | F1 Score | 0.4932 | 0.7540 | 0.7537 | 0.7257 | 0.7231 | 0.7358 | 0.7492 | 0.6154 |
| | T2T | BLEU | 0.3690 | 0.4188 | 0.4117 | 0.3574 | 0.4122 | 0.4061 | 0.4115 | 0.3003 |
| | | ROUGE_1 | 0.7426 | 0.7690 | 0.7661 | 0.7260 | 0.7686 | 0.7552 | 0.7652 | 0.7331 |
| | | ROUGE_L | 0.7081 | 0.7365 | 0.7349 | 0.6914 | 0.7359 | 0.7230 | 0.7328 | 0.6839 |
| | | METEOR | 0.7262 | 0.7452 | 0.7448 | 0.6567 | 0.7493 | 0.7252 | 0.7402 | 0.7111 |

models perform well on CDTP_HP but show weaker results on CDTP_TE, with similar trends observed in QA and T2T tasks. This indicates that the specific requirements of each task and dataset can significantly influence model performance. Consequently, a comprehensive evaluation framework that incorporates multiple tasks and metrics is crucial for fully assessing Chinese LLMs.

**Observation ❷**: **Comparison between QA and KGC tasks.** As illustrated in Figure 2, all Chinese LLMs consistently outperform in KGC compared to QA tasks within the MRR metric. This observation stands in contrast to trends commonly seen in English, where LLMs typically perform better on unstructured QA tasks, largely due to their pretraining on vast amounts of unstructured natural language corpora (Achiam et al., 2023). The divergence highlights the unique linguistic challenges of Chinese—such as polysemy, syntactic ambiguity, and script variation—which can impair model performance on unstructured tasks. In comparison, the structured nature of KGC provides explicit semantic relationships that help alleviate these challenges, resulting in more stable and reliable outcomes. These findings underscore the importance of developing evaluation methodologies that are sensitive to the linguistic characteristics of Chinese.

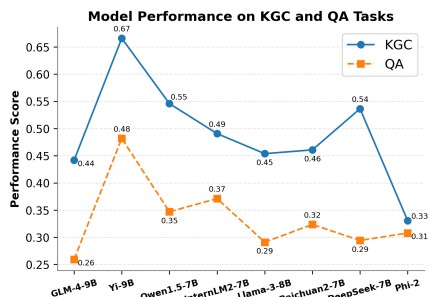

Figure 2: Performance comparison between KGC and QA on CDTP_HP.

**Observation ❸**: **The larger-scale LLMs exhibit better performance.** As illustrated in Figure 3, the performance of base models appears to improve with an increase in model size, especially

for tasks such as T2T and KGC. Larger models, such as Yi-9B and InternLM2-7B, consistently outperform smaller models on these tasks. This observation suggests that increasing the parameters

enhances its ability to capture more complex patterns and semantic relationships, which are crucial for tasks that require deeper language understanding and structured knowledge. However, this trend does not hold as strongly for QA tasks. Despite having a large number of parameters, Yi-9B does not significantly outperform smaller models like DeepSeek-7B in the QA task. This suggests that the QA task, which relies more on open-ended, unstructured questions, may not benefit as directly from an increase in model size, as performance improvement in QA tasks depends more on the model's ability to handle complex reasoning and contextual understanding rather than just the number of parameters. For the ranking experiments of evaluating other base and SFT models, see Appendix D.2.

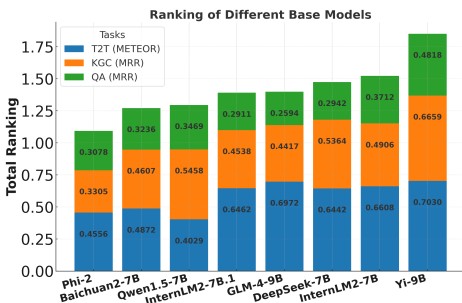

Figure 3: Ranking of 8 base models on CDTP_HP across three tasks. Lager values indicating better performance.

## 5.2 RQ2: SUPERVISED FINE-TUNING (SFT) EXPERIMENTS

**Experiment Design.** We perform Supervised Fine-Tuning (SFT) on these base models and subsequently reassess their performance on these datasets to measure improvements resulting from fine-tuning.

**Experimental Results.** Table 5 and Figure 4 show the performance comparison in terms of three tasks on four datasets after SFT. We have the following key observations.

**Observation ❹: Improvements Across Datasets and Tasks.** As shown in Figure 4, SFT models

consistently outperform their base counterparts on the T2T task across all datasets. This highlights the significant generalization gains introduced by SFT. Notably, Yi-9B and GLM-4-9B exhibit strong and stable generalization capabilities, with their SFT variants achieving consistently high METEOR scores and showing minimal performance degradation across different datasets. This suggests these models possess high adaptability

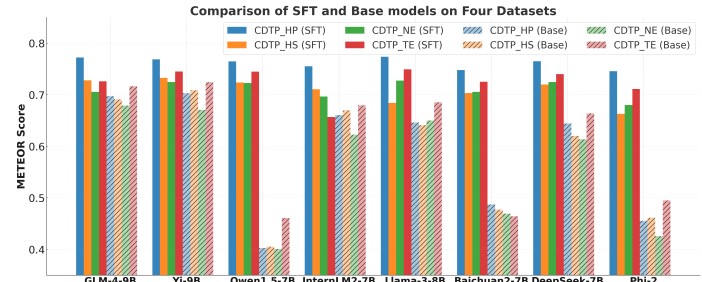

Figure 4: Performance comparison of SFT and Base model on different datasets in T2T task.

to varying linguistic inputs and task nuances, making them reliable across domains. In contrast, models like Phi-2 and Qwen1.5-7B show lower base performance and larger performance gaps between their base and SFT versions, indicating weaker generalization in the absence of fine-tuning. Nevertheless, after SFT, these models demonstrate substantial performance gains, further validating the effectiveness of fine-tuning in enhancing text generation quality. For the effects of other task-based and SFT models, see Appendix D.3 and D.1.

**Observation ❺: Narrowed Performance Gaps After Fine-Tuning.** Fine-tuning improves the performance by reducing the performance gaps between different models, leading to more balanced results across multiple tasks. Specifically, fine-tuning enhances generalization and stability, allowing models of various sizes and architectures to perform more consistently. After fine-tuning, most models approach near-perfect scores, indicating that the fine-tuning data is both broad and high-quality, making it applicable to a wide range of tasks and model types. These findings emphasize the significance of our dataset in optimizing model performance across different tasks.

## 5.3 RQ3: ROBUSTNESS UNDER OUT-OF-DISTRIBUTION (OOD) DATA

**Experiment Design.** We conduct the robustness experiments across three tasks with specialized datasets to verify the robustness of Chinese LLMs Yi-9B on OOD data (Liu et al., 2023), before and

after SFT using our proposed data. For the KGC task, we select YAGO3-10 (Wang et al., 2024c) as the OOD dataset and evaluate performance using the MRR metric. For the QA task, HotpotQA (Yang et al., 2018) is employed, with evaluation based on the F1 Score. For the T2T task, we adopt the WebNLG dataset (Gardent et al., 2017), using METEOR as the evaluation metric.

**Experimental Results.** Figure 5 presents the performance comparison across three tasks on three OOD datasets, before and after SFT. We have the following key observations.

**Observation ➏: Improved Performance After Fine-Tuning.** As illustrated in Figure 5, SFT significantly enhances model performance across all tasks (KGC, QA, T2T) when tested with OOD data. For example, in the QA task, the SFT model achieves a notable improvement over the base model, indicating enhanced reasoning capabilities under distributional shifts. Similar trends are observed in KGC and T2T tasks, where SFT helps the models generalize better to unseen entities and text patterns. These results validate that Chinese LLMs possess a certain level of inherent robustness when facing OOD data. More importantly, the consistent performance gains after SFT further demonstrate the effectiveness of our proposed CDTP datasets.

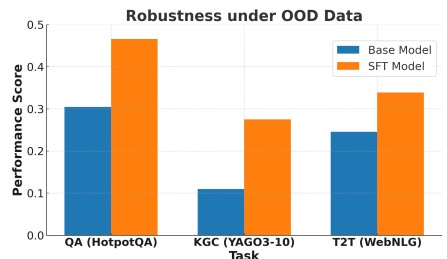

Figure 5: Comparison of robustness between base models and SFT models under the out-of-distribution (OOD) data.

**Observation ➐: Task-Specific Robustness.** While SFT consistently enhances performance across all OOD tasks, the extent and nature of the improvements vary by task type. The QA task exhibits the most substantial gains, reflecting improved reasoning and contextual understanding. KGC, despite its lower baseline, benefits markedly from SFT, suggesting enhanced generalization of structured knowledge. In contrast, T2T demonstrates more moderate yet stable improvements, indicating better alignment in text generation. These task-specific variations highlight that robustness gains are not uniform and underscore the necessity of targeted fine-tuning strategies to adapt Chinese LLMs effectively to diverse real-world scenarios.

## 6 CONCLUSION AND FUTURE DIRECTIONS

In this paper, we propose a **C**omprehensive **B**enchmark for **E**valuating **C**hinese **L**arge **L**anguage **M**odels (CB-ECLLM) based on the newly constructed Chinese Data-Text Pair (CDTP) dataset. Specifically, CDTP comprises over 7 million meticulously aligned text pairs, each consisting of unstructured text coupled with one or more corresponding triples, alongside a total of 15 million triples spanning four critical domains: History and Politics, Humanities and Society, Technology and Economics, and Nature and Environment. CDTP supports tasks like KGC, T2T, and QA, with a special emphasis on the unique linguistic challenges of Chinese. Furthermore, we conduct rigorous evaluations through extensive experiments and ablation studies to comprehensively assess the effectiveness, Supervised Fine-Tuning (SFT), and robustness of the benchmark. This dataset offers a comprehensive benchmark for evaluating reasoning, generalization, and structured knowledge comprehension in Chinese LLMs, paving the way for AGI. One limitation is the proposed CDTP dataset only contains textual inputs and structured triples, without incorporating cross-modal signals such as images or audio, which constrains its applicability in multimodal settings.

Building on these promising results and limitations (Details of limitations see Appendix E), we outline critical research directions to address remaining challenges:

- **Enhanced Domain and Entity Coverage.** We plan to expand the CDTP dataset by incorporating data from more diverse sources, including underrepresented domains and long-tail entities. This will improve model robustness and adaptability in specialized or low-resource scenarios.

- **Cross-Domain Generalization.** Future work will explore domain-adaptive pretraining and meta-learning approaches to enhance generalization across unseen domains and heterogeneous data modalities, such as tabular, visual, or conversational inputs.

- **Linguistic and Cultural Enrichment.** Future work will extend CDTP beyond factual triples by incorporating linguistically and culturally rich phenomena, including metaphorical expressions, pragmatic markers, and culturally embedded references.

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

## A    USE OF LARGE LANGUAGE MODELS

We employed LLMs exclusively as editing assistants to enhance grammar, clarity, and conciseness of the manuscript. All technical contributions, including experimental design, data processing, evaluation, and conclusions, were conceived, implemented, and validated by the human authors. Edits suggested by LLMs were carefully reviewed and either accepted or modified by the authors; no numerical results, figures, or analyses were generated or approved solely by the LLM.

## B    DATASET

### B.1    COMPARISON OF CHINESE DATASETS

Table 6 compares four key Chinese datasets. CDTP is unique in its focus on structured knowledge evaluation through triple–text pairs, making it particularly well-suited for tasks like Knowledge Graph Completion (KGC), Question Answering (QA), and Triple to Text Generation (T2T). In comparison, C-Eval (Huang et al., 2023) and CMMLU (Li et al., 2023) are centered around general knowledge and multiple-choice evaluation, focusing on reasoning and broader knowledge acquisition. While each dataset serves distinct evaluation purposes, CDTP excels in testing knowledge-grounded reasoning specifically for Chinese LLMs.

Table 6: Comparison of Chinese Datasets

| Dataset | Scale | Key Focus |
|---|---|---|
| CDTP (Ours) | 7M+ pairs, 15M+ triples | Structured knowledge evaluation for Chinese LLMs |
| C-Eval | 13.9K questions | General knowledge, multiple-choice evaluation |
| CMMLU | 12K questions | General knowledge, multiple-choice evaluation |

### B.2    TOP 20 RELATIONS

Knowledge graphs (KGs) are inherently long-tailed, with a small number of relations occurring very frequently while the majority appear only sparsely. Such skewness can introduce noise and reduce the reliability of downstream evaluation. To mitigate this effect, we analyzed the frequency distribution of relations in the triples and retained only high-frequency ones. This filtering step emphasizes commonly occurring and semantically meaningful relations while suppressing rare or noisy edges. Figure 6 illustrates the top 20 most frequent relations in the dataset.

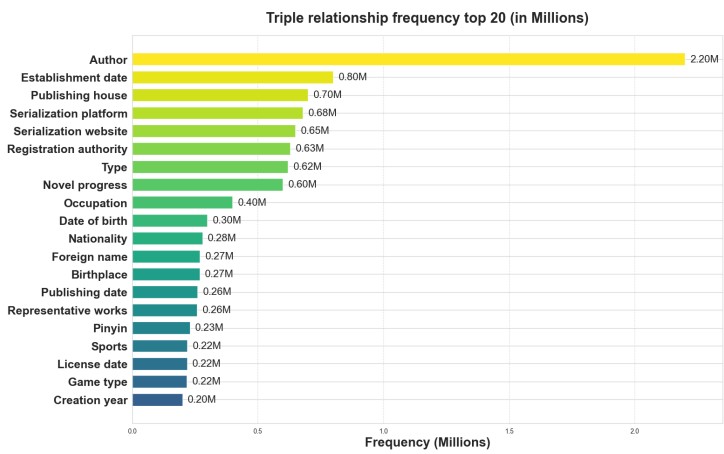

Figure 6: Triple relationship frequency top 20.

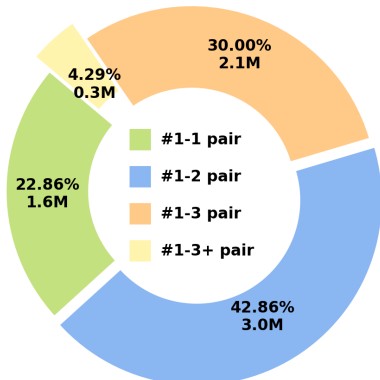

Figure 7: Distribution of data pairs by the number of associated relation triples. A **1-1 pair** indicates one text aligned with a single triple; **1-2** and **1-3** pairs follow the same convention. The **1-3+** category denotes cases where one text is aligned with more than three triples.

### B.3 DATA STATISTICS

The proposed CDTP dataset is organized into data pairs, each consisting of a main entity, a set of relation triples, and a corresponding text description. The number of triples associated with each text ranges from 1 to 14, and the description is designed to comprehensively capture the information expressed by the triples. To examine the distribution of triples per data pair, we conduct a statistical analysis, summarized in Figure 7.

The results reveal that the dataset is dominated by pairs containing no more than three triples, which account for roughly 96% of all entries. Among these, the **1-2 pair** (one text aligned with two triples) is the most frequent, contributing about 3 million instances. In contrast, pairs with higher triple counts are rare; for example, only three **1-14 pairs** (one text aligned with fourteen triples) are observed, underscoring the sparsity of such cases.

Given this skewed distribution, our analysis and downstream experiments primarily focus on **1-3+** pairs. This choice not only simplifies the presentation but also reflects the dataset's practical utility, as most real-world applications involve relatively few relation triples per text.

## C MODEL DETAILS APPENDIX

### C.1 DETAILS OF EVALUATION MODELS

To comprehensively evaluate the performance of large language models (LLMs) on our proposed dataset, we selected eight state-of-the-art models across diverse architectures and training paradigms. These models represent a mix of general-purpose and Chinese-focused LLMs, providing a robust benchmark for comparison. The evaluated models are as follows:

- **GLM-4-9B** [1]: A highly capable general-purpose LLM optimized for multiple languages and tasks,supporting 26 languages.
- **Yi-9B** [2]: A Chinese-focused, high-quality natural language understanding and generation model designed as a bilingual language model, trained on a 3T multilingual corpus, and excelling in coding and math within the Yi series of models.
- **Qwen1.5-7B** [3]: A powerful and efficient model, designed for reasoning and text generation tasks, featuring substantial improvements in aligning chat models with human preferences and allowing for the handling of long contexts.

---

[1]Available at: huggingface.co/THUDM/glm-4-9b
[2]Available at: huggingface.co/01-ai/Yi-9B
[3]Available at: huggingface.co/Qwen/Qwen1.5-7B

- **InternLM2-7B** [4]: The latest iteration of the InternLM series is optimized for Chinese-centric datasets and offers effective support for ultra-long contexts of up to 200,000 characters.

- **Llama-3-8B** [5]: The Llama 3 instruction-tuned models, recognized as a leading general-purpose LLM for multilingual applications, are specifically optimized for dialogue use cases, all while emphasizing helpfulness and safety in their design.

- **Baichuan2-7B** [6]: A model focused on the Chinese language, demonstrates exceptional capabilities in text completion and knowledge-intensive tasks, having attained the highest performance on various authoritative benchmarks in Chinese, English, and multilingual general and domain-specific domains.

- **DeepSeek-7B** [7]: Known for its ability to handle complex reasoning and domain-specific queries, this model places in the top 3 on AlignBench, ranks as a top-tier performer on MT-Bench, specializes in math, code, and reasoning, and supports 128K context length.

- **Phi-2** [8]: A next-generation LLM designed for efficiency and high-quality text generation, augmented with a new data source comprising various NLP synthetic texts and curated websites to enhance safety and educational value.

- **Qwen3-32B**[9]: A latest-generation model in the Qwen series, representing a 32B-parameter variant within a comprehensive suite of dense and mixture-of-experts (MoE) LLMs.

- **GPT-4**[10]: A high-capacity large language model developed by OpenAI, widely used as a strong general-purpose baseline for challenging reasoning and knowledge-intensive tasks.

- **Claude-sonnet-4.5**[11]: A mid-sized model in the Claude family that balances capability and efficiency, supporting advanced reasoning, coding, and tool-use scenarios with a long context window.

- **Gemini-2.5-flash**[12]: A Gemini-series model optimized for price–performance, providing fast, low-latency inference while maintaining competitive performance on large-scale language understanding and generation tasks.

These models were chosen to ensure a diverse evaluation across different architectural designs and training objectives. By evaluating their performance across the 12 tasks, derived from combining four dataset categories with three distinct task types, we aim to gain insights into the strengths, limitations, and task-specific capabilities of each model. This comprehensive evaluation provides valuable benchmarks for future model development and research in LLM performance optimization.

## C.2 TASK IMPLEMENTATION DETAILS

To effectively evaluate Chinese LLMs, we transform the CDTP dataset into formats tailored for specific tasks. For constructing challenging single-choice questions (one gold answer and nine distractors), we first build a candidate pool for each gold entity by combining its 1-hop and 2-hop neighbors in KGs with entities from the same type and similar corpus-frequency bins. A semantic-feasibility filter using a pretrained Chinese sentence-embedding model is then applied to remove near-synonyms of the gold answer and candidates that are semantically unrelated to the question context.

Distractors are selected based on a heuristic that prioritizes entity-type consistency, high semantic confusability, and coverage across frequency/subdomain to avoid homogeneous sets. The ten choices are then shuffled per item using a pseudorandom permutation for evaluation.

To specifically address Chinese language challenges, such as polysemy and ambiguity, we include distractors derived from KG neighbors and same-type entities that are semantically confusable with

---

[4]Available at: huggingface.co/internlm/internlm2-7b

[5]Available at: huggingface.co/meta-llama/Meta-Llama-3-8B

[6]Available at: huggingface.co/baichuan-inc/Baichuan2-7B-Base

[7]Available at: huggingface.co/deepseek-ai/deepseek-llm-7b-base

[8]Available at: huggingface.co/microsoft/phi-2B

[9]Available at: https://huggingface.co/Qwen/Qwen3-32B

[10]Available at: https://openai.com/index/gpt-4

[11]Available at: https://www.anthropic.com/claude

[12]Available at: https://deepmind.google/models/gemini

different senses of the target token. This targeted construction allows us to assess model robustness under Chinese semantic ambiguity within the same standardized evaluation framework.

The downstream tasks are employed as follows:

- **Question Answering (QA)**: The QA task involves generating questions based on two components of a triple (head entity, relation, or tail entity) and predicting the third component as the answer. For example, given the triple "(±)反式菊酸, 分子式, C10H16O2"((±) trans-chrysanthemic acid,molecular formula,C10H16O2), the model is asked "What is the molecular formula of "(±)反式菊酸" ((±) trans-chrysanthemic acid?)", and select the most appropriate option from ten choices as shown in Table 7.

Table 7: The example of the QA tasks.

| Query | (±)反式菊酸的分子式是什么? 
 What is the molecular formula of "(±) trans-chrysanthemic acid"? |
|---|---|
| Options | A. C9H16O2, **B. C10H16O2**, C. C11H16O2, D. C11H16O3, E. C12H18O2, F. C10H15O2, G. C10H16O3, H. C9H15O2, I. C11H15O2, J. C10H17O2. |

**Knowledge Graph Completion (KGC)**: The KGC task evaluates a model's ability to predict the missing element of an incomplete triple. For instance, given an incomplete triple (Aisin-Gioro Xibao, date of birth, ?) "(爱新觉罗·锡保, 出生日期, ?)" , the model is required to choose the option that best fits the missing part of the triple from a set of options as shown in Table 8.

Table 8: The example of the KGC tasks.

| Triple | (爱新觉罗·锡保, 出生日期, ?) 
 (Aisin-Gioro Xibao, date of birth, ?) |
|---|---|
| Options | A. 1680, **B. 1681**, C. 1682, D. 1683, E. 1684, F. 1685, G. 1686, H. 1687, I. 1688, J. 1689. |

- **Triple-to-Text Generation (T2T)**: In the T2T task, the model generates a coherent and fluent natural language description based on a given triple, as shown in Table 9.

Table 9: The example of the T2T tasks.

| Triples | ["延庆宫, 作者, 勾台符", "延庆宫, 创作年代, 宋代"] 
 ["Yanqing Palace, author, Gou Taifu", "Yanqing Palace, creation period, Song Dynasty"] |
|---|---|
| Text | 《延庆宫》是宋代诗人勾台符的作品之一。 
 "Yanqing Palace" is one of the works of Gou Taifu, a poet of the Song Dynasty. |

C.3 DETAILS OF EVALUATION METRICS

To comprehensively evaluate the performance of the models across the three tasks, different metrics are employed based on the specific characteristics of each task:

**Question Answering (QA) Task:** For the QA task, the following evaluation metrics are employed:

- *MRR (Mean Reciprocal Rank)*: Computes the mean of the reciprocal ranks of the first correct answer across all queries. A higher MRR value indicates superior performance in ranking the correct answer earlier in the result list.
- *ACC (Accuracy)*: Denotes the proportion of questions for which the model produces the correct answer, serving as a direct measure of prediction correctness.

- *F1 Score*: Represents the harmonic mean of precision and recall, offering a balanced evaluation that accounts for both false positives and false negatives in classification.

**Knowledge Graph Completion (KGC) Task:**   This task assesses the model's capability to infer missing entities or relations within knowledge graph triples. The evaluation metrics include:

- *MRR (Mean Reciprocal Rank)* and *F1 Score*: These metrics, also utilized in the QA task, are used to evaluate the ranking quality of the predicted entities and the overall classification performance, respectively.

- *Hits@k*: Measures the proportion of cases where the correct entity appears within the top-$k$ ranked candidates. This metric provides insight into the model's effectiveness in prioritizing correct predictions.

**Triple-to-Text (T2T) Task:**   The T2T task focuses on generating coherent and semantically accurate natural language descriptions from structured knowledge graph triples. The following automatic evaluation metrics are adopted:

- *BLEU (Bilingual Evaluation Understudy)*: Assesses the degree of n-gram overlap between the generated text and reference text, and is widely used in machine translation and text generation tasks.

- *ROUGE (Recall-Oriented Understudy for Gisting Evaluation)*: Evaluates the overlap of unigrams, bigrams, and longer sequences between the generated and reference texts, primarily measuring recall-based performance.

- *METEOR (Metric for Evaluation of Translation with Explicit ORdering)*: Incorporates precision, recall, stemming, synonym matching, and word order to provide a more comprehensive assessment of textual similarity and fluency.

The specific calculation formulas for each evaluation metric are presented below:

- **MRR (Mean Reciprocal Rank):** MRR evaluates the average reciprocal rank of the correct answers:

$$\text{MRR} = \frac{1}{N} \sum_{i=1}^{N} \frac{1}{\text{rank}_i},$$

  where $N$ is the total number of queries, and $\text{rank}_i$ denotes the rank position of the first correct answer for the $i$-th query. If the correct answer is not found in the returned results, $\frac{1}{\text{rank}_i}$ is treated as 0.

- **ACC (Accuracy):** Accuracy is a straightforward metric that measures the proportion of correctly answered questions over the total number of questions:

$$\text{Accuracy} = \frac{N_{\text{correct}}}{N_{\text{total}}}$$

  where $N_{\text{correct}}$ denotes the number of questions for which the model predicted the correct answer, and $N_{\text{total}}$ is the total number of questions. In QA tasks, a prediction is considered correct if it exactly matches the ground truth answer.

- **F1 Score:** The F1 Score is the harmonic mean of precision and recall, providing a single metric that balances both false positives and false negatives:

$$\text{F1 Score} = 2 \times \frac{\text{Precision} \times \text{Recall}}{\text{Precision} + \text{Recall}},$$

  where $\text{Precision} = \frac{\text{True Positives}}{\text{True Positives} + \text{False Positives}}$ is how many of the samples predicted to be positive are true positive samples; $\text{Recall} = \frac{\text{True Positives}}{\text{True Positives} + \text{False Negatives}}$ is how many of the real positive class samples were successfully predicted to be positive classes.

- **Hits@k:** Hits@k measures the proportion of correct answers that appear within the top-$k$ predicted results:

$$\text{Hits@}k = \frac{1}{N} \sum_{i=1}^{N} \mathbb{I}\left(\text{rank}_i \leq k\right),$$

where $\mathbb{I}(\cdot)$ is the indicator function that returns 1 if the condition is true, and 0 otherwise. A higher Hits@k indicates better recall among the top-$k$ predictions.

- **BLEU (Bilingual Evaluation Understudy):** BLEU is a widely used metric for evaluating the quality of machine-generated text by comparing n-gram overlap between the generated text and reference texts. It is particularly effective for tasks like machine translation and text generation, where word order and n-gram similarity are important:

$$\text{BLEU} = \exp\left(\min\left(1, \frac{\text{count}_{\text{match}}(n)}{\text{count}_{\text{generated}}(n)}\right) \cdot \text{BP}\right)$$

Here, $\text{count}_{\text{match}}(n)$ refers to the number of matching n-grams between the generated and reference texts, $\text{count}_{\text{generated}}(n)$ is the total number of n-grams in the generated text, and BP is the brevity penalty that penalizes overly short translations. BLEU is typically calculated using unigram, bigram, and higher-order n-grams.

- **ROUGE (Recall-Oriented Understudy for Gisting Evaluation):** ROUGE measures the recall-based overlap between n-grams, word sequences, or word pairs in the generated text and reference texts. It is commonly used for evaluating summarization and text generation tasks. The most common variants are ROUGE-N (measuring n-gram overlap) and ROUGE-L (measuring longest common subsequence overlap):

$$\text{ROUGE} - \text{N} = \frac{\sum_{n=1}^{N} \text{count}_{\text{match}}(n)}{\sum_{n=1}^{N} \text{count}_{\text{reference}}(n)}$$

In the case of ROUGE-L, the score is computed based on the longest common subsequence (LCS) between the generated and reference texts, reflecting the fluency and coherence of the generated content.

- **METEOR (Metric for Evaluation of Translation with Explicit ORdering):**

$$\text{METEOR} = \frac{1}{n} \sum_{i=1}^{n} \frac{\text{precision}_i \cdot \text{recall}_i}{\alpha \cdot \text{precision}_i + (1 - \alpha) \cdot \text{recall}_i},$$

where $n$ is the number of words matched and $\alpha$ is a modulating parameter (usually 0.9).

# D EXPERIMENTS

## D.1 EXPERIMENTS ON LLMS WITH DIFFERENT PARAMETER SIZES

**Experimental Design.** We conduct a comprehensive evaluation of two Chinese LLMs—Qwen1.5-7B and Qwen3-32B—in both their base and Supervised Fine-Tuning (SFT) variants, together with three closed-source LLMs: GPT-4, Claude-sonnet-4.5, and Gemini-2.5-flash. The models are assessed on three knowledge-driven tasks—Knowledge Graph Completion (KGC), Question Answering (QA), and Triple-to-Text Generation (T2T)—across four domain-specific datasets.

**Model Selection.** We evaluate four Qwen variants: Qwen1.5-7B-Base and Qwen3-32B-Base (base models with 7B and 32B parameters), together with their supervised fine-tuned counterparts, Qwen1.5-7B-SFT and Qwen3-32B-SFT. Additionally, we conduct additional experiments with prominent multilingual models: GPT-4, Claude-sonnet-4.5, and Gemini-2.5-flash.

**Experimental Results and Analysis.** Table 10 presents the comprehensive results. Our analysis yields the following key findings:

Table 10: LLMs with different parameter sizes Experiments

| Datasets | Tasks | Metrics | Qwen3-32B-Base | Qwen1.5-7B-Base | Qwen1.5-7B-SFT | Qwen3-32B-SFT | GPT-4 | Claude-sonnet-4.5 | Gemini-2.5-flash |
|---|---|---|---|---|---|---|---|---|---|
| CDTP_HP | KGC | MRR | 0.7004 | 0.5458 | 0.7877 | 0.8140 | 0.7244 | 0.6724 | 0.6943 |
| | | Hits@1 | 0.5645 | 0.4005 | 0.7044 | 0.7375 | 0.6024 | 0.5340 | 0.5620 |
| | | F1 Score | 0.7216 | 0.5719 | 0.8266 | 0.8489 | 0.7518 | 0.6962 | 0.7196 |
| | QA | MRR | 0.5308 | 0.3469 | 0.6083 | 0.6448 | 0.5563 | 0.4896 | 0.5192 |
| | | ACC | 0.4260 | 0.2425 | 0.5525 | 0.6033 | 0.4678 | 0.3846 | 0.4100 |
| | | F1 Score | 0.5975 | 0.3903 | 0.7118 | 0.7526 | 0.6374 | 0.5555 | 0.5816 |
| | T2T | BLEU | 0.5875 | 0.1471 | 0.4672 | 0.7088 | 0.5772 | 0.5537 | 0.5850 |
| | | ROUGE_1 | 0.7163 | 0.5516 | 0.7859 | 0.7957 | 0.7050 | 0.6825 | 0.7205 |
| | | ROUGE_L | 0.6594 | 0.4969 | 0.7551 | 0.7663 | 0.6434 | 0.6219 | 0.6593 |
| | | METEOR | 0.7060 | 0.4029 | 0.7647 | 0.7834 | 0.6785 | 0.6437 | 0.7122 |
| CDTP_HS | KGC | MRR | 0.7017 | 0.5363 | 0.7966 | 0.8068 | 0.7371 | 0.6957 | 0.7087 |
| | | Hits@1 | 0.5655 | 0.3975 | 0.7140 | 0.7282 | 0.6238 | 0.5680 | 0.5845 |
| | | F1 Score | 0.7225 | 0.5689 | 0.8331 | 0.8427 | 0.7684 | 0.7245 | 0.7378 |
| | QA | MRR | 0.5279 | 0.3072 | 0.6437 | 0.6640 | 0.5698 | 0.5220 | 0.5306 |
| | | ACC | 0.4147 | 0.1960 | 0.5858 | 0.6101 | 0.4683 | 0.4114 | 0.4100 |
| | | F1 Score | 0.5863 | 0.3278 | 0.7388 | 0.7578 | 0.6379 | 0.5830 | 0.5816 |
| | T2T | BLEU | 0.5879 | 0.1361 | 0.3735 | 0.6678 | 0.5885 | 0.5576 | 0.5874 |
| | | ROUGE_1 | 0.7129 | 0.5527 | 0.7513 | 0.7659 | 0.7101 | 0.6845 | 0.7221 |
| | | ROUGE_L | 0.6486 | 0.5025 | 0.7109 | 0.7289 | 0.6493 | 0.6248 | 0.6601 |
| | | METEOR | 0.7017 | 0.4052 | 0.7239 | 0.7495 | 0.6906 | 0.6551 | 0.7212 |
| CDTP_NE | KGC | MRR | 0.7808 | 0.6002 | 0.8245 | 0.8416 | 0.7922 | 0.7500 | 0.7710 |
| | | Hits@1 | 0.6855 | 0.4735 | 0.7590 | 0.7870 | 0.6981 | 0.6425 | 0.6680 |
| | | F1 Score | 0.8134 | 0.6427 | 0.8630 | 0.8808 | 0.8222 | 0.7823 | 0.8010 |
| | QA | MRR | 0.5951 | 0.3149 | 0.6284 | 0.6370 | 0.6188 | 0.5504 | 0.5806 |
| | | ACC | 0.5175 | 0.2320 | 0.5853 | 0.6030 | 0.5486 | 0.4617 | 0.4940 |
| | | F1 Score | 0.6820 | 0.3766 | 0.7384 | 0.7523 | 0.7085 | 0.6318 | 0.6613 |
| | T2T | BLEU | 0.5375 | 0.1213 | 0.4033 | 0.6678 | 0.5434 | 0.5128 | 0.5504 |
| | | ROUGE_1 | 0.6781 | 0.5399 | 0.7481 | 0.7602 | 0.6703 | 0.6511 | 0.6897 |
| | | ROUGE_L | 0.6403 | 0.5029 | 0.7249 | 0.7373 | 0.6309 | 0.6093 | 0.6529 |
| | | METEOR | 0.6606 | 0.4008 | 0.7226 | 0.7431 | 0.6376 | 0.6133 | 0.6813 |
| CDTP_TE | KGC | MRR | 0.7201 | 0.4973 | 0.7856 | 0.8145 | 0.7220 | 0.6829 | 0.7033 |
| | | Hits@1 | 0.6003 | 0.3620 | 0.6965 | 0.7433 | 0.6064 | 0.5465 | 0.5735 |
| | | F1 Score | 0.7502 | 0.5316 | 0.8211 | 0.8527 | 0.7550 | 0.7068 | 0.7289 |
| | QA | MRR | 0.5554 | 0.3486 | 0.6563 | 0.6637 | 0.5934 | 0.5377 | 0.5565 |
| | | ACC | 0.4520 | 0.2425 | 0.6048 | 0.6205 | 0.5000 | 0.4320 | 0.4510 |
| | | F1 Score | 0.6226 | 0.3903 | 0.7537 | 0.7658 | 0.6667 | 0.6034 | 0.6216 |
| | T2T | BLEU | 0.5943 | 0.1665 | 0.4117 | 0.6769 | 0.5936 | 0.5693 | 0.5938 |
| | | ROUGE_1 | 0.7203 | 0.5875 | 0.7661 | 0.7754 | 0.7193 | 0.6978 | 0.7286 |
| | | ROUGE_L | 0.6645 | 0.5364 | 0.7349 | 0.7423 | 0.6643 | 0.6419 | 0.6719 |
| | | METEOR | 0.7113 | 0.4608 | 0.7448 | 0.7594 | 0.7038 | 0.6734 | 0.7332 |

1. **Effectiveness of SFT**: Supervised fine-tuning (SFT) consistently boosts performance across all four datasets and three task types (KGC, QA, and T2T). Both 7B and 32B SFT models show clear gains over their base counterparts.

2. **Performance Ranking**: The overall hierarchy of model performance is: Qwen3-32B-SFT > Qwen1.5-7B-SFT > GPT 4> Qwen3-32B-Base > Gemini-2.5-flash > Claude-sonnet-4.5 > Qwen1.5-7B-Base. This shows that increased parameter size alone does not guarantee strong performance. The Qwen3-32B-Base only modestly improves over smaller or closed-source systems, and is still clearly outperformed by GPT-4, whereas both SFT variants consistently surpass GPT-4 across three tasks. This pattern indicates that task- and language-specific fine-tuning is the dominant factor in realizing the benefits of a larger-capacity model, and that a well-aligned 7B model can not only close the gap with much larger proprietary LLMs but, in our setting, even surpass them.

3. **Task-Specific Gains**: Performance improvements from SFT are systematically larger in KGC than in QA tasks, suggesting that structured knowledge completion benefits more substantially from fine-tuning than open-ended question answering.

4. **Scalability of Larger Models**: The Qwen3-32B-SFT achieves the strongest overall results, demonstrating that larger models, when properly fine-tuned, can deliver superior capabilities beyond the reach of smaller models.

## D.2  RANKING OF DIFFERENT BASE AND SFT MODELS

**Experiment Design.** We comprehensively rank the performance of eight base and SFT Chinese LLMs across three tasks—Knowledge Graph Completion (KGC), Question Answer (QA), and Triple to Text Generation (T2T)—using four distinct datasets.

**Experimental Results.** Figure 8 and Figure 9 provide the bar graphs to illustrate the performance rankings of Chinese LLMs with respect to their parameter sizes. We have the following observations.

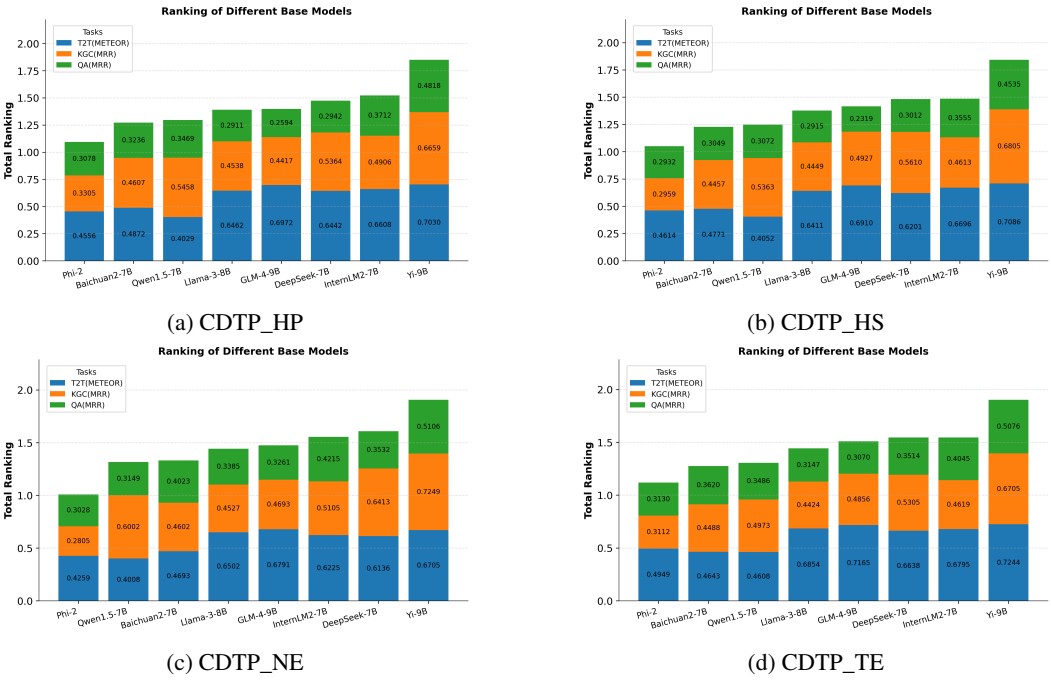

Figure 8: Ranking of 8 base models on (a) CDTP_HP & (b) CDTP_HS & (c) CDTP_NE & (d) CDTP_TE across three tasks. Lager values indicating better performance.

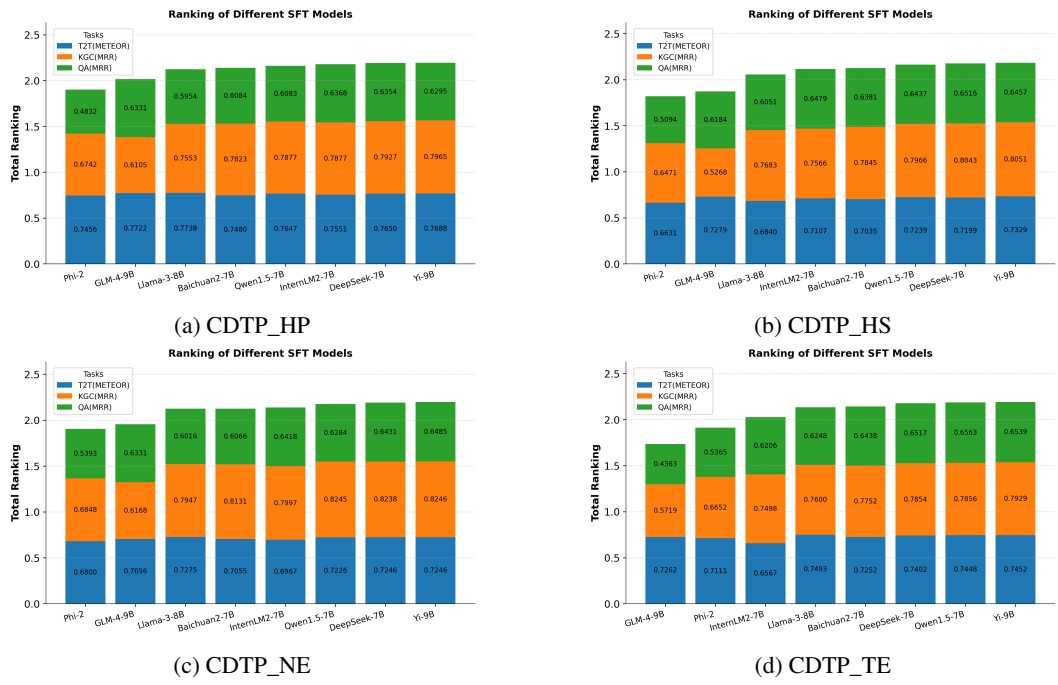

Figure 9: Ranking of 8 SFT models on (a) CDTP_HP & (b) CDTP_HS & (c) CDTP_NE & (d) CDTP_TE across three tasks. Lager values indicating better performance.

**Observation ❽**: **The larger-scale LLMs exhibit better performance.** As shown in Figure 8, the performance of the eight base models generally improves with the increase in model size. Notably,

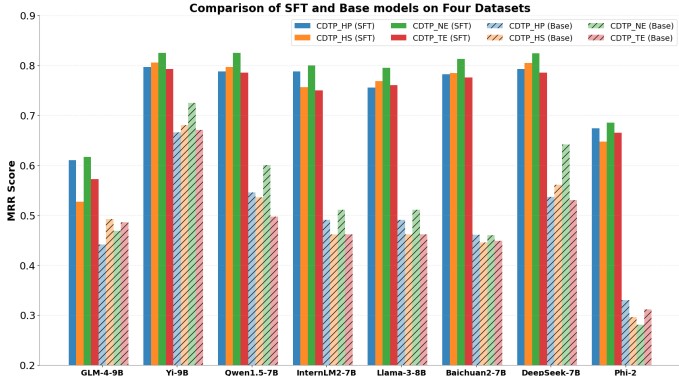

(a) Performance comparison of SFT and Base model on different datasets in KGC task.

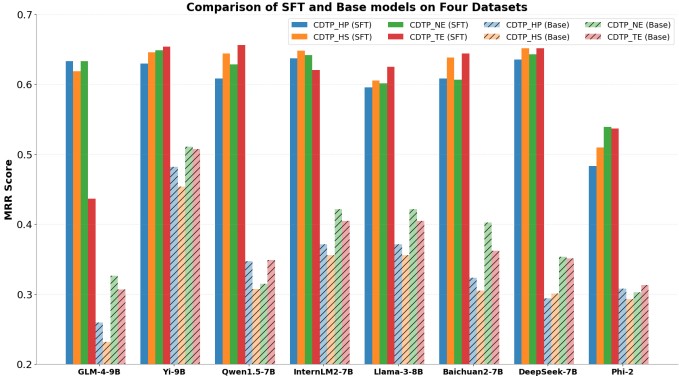

(b) Performance comparison of SFT and Base model on different datasets in QA task.

Figure 10: Performance comparison of SFT and Base model on different datasets in KGC and QA tasks.

models with a larger number of parameters, such as Yi-9B, consistently outperform smaller-scale models across the three benchmark tasks. This trend suggests that increasing parameter count can effectively enhance a model's capacity to capture complex patterns and semantic relationships. However, as previously analyzed, this effect is less pronounced in the QA task, which primarily involves open-ended and unstructured questions. For such tasks, performance relies more heavily on the model's abilities in complex reasoning and contextual understanding rather than on parameter size alone.

**Observation ⑨**: **Substantial Enhancement After SFT.** As illustrated in Figure 9, all SFT LLMs demonstrate improved performance compared to their corresponding base versions. However, unlike in the base models, the performance gains from increasing parameter size are less significant across the three tasks. Overall, SFT LLMs exhibit more stable and consistent performance, with models of similar sizes achieving comparable results after fine-tuning on the same dataset. This indicates that supervised fine-tuning can mitigate performance differences arising from parameter scale to some extent, leading to convergence in task performance among similarly sized models.

### D.3 SUPERVISED FINE-TUNING (SFT) EXPERIMENTS

**Experiment Design.** We perform Supervised Fine-Tuning (SFT) on these base models and subsequently reassess their performance on these datasets to measure improvements resulting from fine-tuning.

**Experimental Results.** Table 5 and Figure 10 show the performance comparison in terms of three tasks on four datasets after SFT. We have the following key observations.

**Observation ⑩: Improvements Across Datasets and Tasks.** As shown in Figure 10, SFT LLMs demonstrate consistent superiority over base LLMs across all four datasets (CDTP_HP/HS/NE/TE) in both KGC and QA tasks, with GLM-4-9B and Yi-9B achieving the highest MRR scores while maintaining minimal performance variance, indicating robust cross-domain adaptability. In contrast, smaller models (Phi-2, Qwen1.5-7B) exhibit significant SFT-induced improvements, revealing their stronger dependency on task-specific fine-tuning. The performance gaps between base and SFT versions are systematically larger in KGC tasks than in QA tasks, suggesting that structured knowledge completion benefits more substantially from supervised fine-tuning compared to open-ended question answering.

## E    LIMITATIONS

While our work significantly advances Chinese data-to-text generation, it also presents several limitations:

- **Domain and Entity Coverage.** Although the CDTP dataset is large and high-quality, its coverage is constrained by the inherent biases of the source data. Underrepresented domains and infrequent entities may not be adequately captured, potentially limiting model performance on specialized or low-resource tasks.

- **Domain Generalization.** CDTP spans four major categories, but achieving comprehensive domain diversity remains challenging. While models fine-tuned on CDTP perform well on in-domain and selected out-of-distribution tasks, their ability to generalize to entirely novel domains or modalities is yet to be fully validated.

- **Linguistic and Cultural Nuance.** The dataset focuses primarily on factual knowledge expressed through structured triples, which limits its capacity to evaluate deeper linguistic competencies such as metaphorical reasoning, pragmatic inference, or cultural context understanding. Although CDTP addresses morphological ambiguity and polysemy through precise triple-text alignment, it does not explicitly model sociolinguistic or culturally embedded phenomena—factors essential for comprehensive evaluation of advanced Chinese language understanding.

