# OpenReview forum: "CDTP: A Large-Scale Chinese Data-Text Pair Dataset for Comprehensive Evaluation of Chinese LLMs"
_ICLR.cc/2026/Conference — ICLR 2026 Conference Desk Rejected Submission_

### Official Review · Reviewer_YJ4r · 2025-10-25

**Soundness:** 1
**Presentation:** 3
**Contribution:** 1
**Rating:** 4
**Confidence:** 5

**Summary:**

This paper introduces CDTP, a dataset of aligned Chinese text–triple pairs. It also porpose a CB-ECLLM benchmark to evaluate three representative tasks: Knowledge Graph Completion (KGC), Triple-to-Text Generation (T2T), and Question Answering (QA). The experiment evaluates 8 LLMs to assess effectiveness, SFT gains, and OOD robustness.

**Strengths:**

1. The dataset is large, it might be useful for Chinese NLP community.

2. The paper provides a detailed data construction process, and promises to open source code and data to facilitate reproducibility.

**Weaknesses:**

1. The paper claims that Chinese language poses a challenge to LLM (L46), so this dataset was constructed. However, there is no discussion on why this dataset can help Chinese LLM training. The example in Tab. 1 could be reproduced in alternative languages.

2. L245 `we selected eight state-of-the-art models'. These models are not representative because they only include 7-9b open source models and lack other scale and closed source commercial models as baseline. Also, these models are not state-of-the-art, some of them are published in 2023.

3. The effectiveness of the evaluation tasks. The paper proposes three tasks: KGC, T2T, and QA, but does not discuss the justification for these task designs. For example, why should LLMs be evaluated using triples instead of natural language? These tasks also fail to represent the challenges LLM faces when constructing knowledge graphs. For example, in the example L805, only a person's name is provided and the model is asked to complete the birth date, while the options include different years. This question does not clarify the person being referred to and cannot constitute a valid evaluation. If this question provides more context in the form of natural language, it seems to be more effective.

**Questions:**

See `Weakness' section.

---

> ### Author Response · Authors · 2025-11-20
> **Reply for Reviewer YJ4r**
>
> ``W.1:`` The paper claims that Chinese language poses a challenge to LLM (L46), so this dataset was constructed. However, there is no discussion on why this dataset can help Chinese LLM training. The example in Tab. 1 could be reproduced in alternative languages.
>
> ``Re:`` We agree that the previous version did not make the connection between Chinese-specific challenges and the design of CDTP sufficiently explicit.
>
> * The example in Tab. 1 could be reproduced in alternative languages?
> >Table 1 is not intended to claim that polysemy itself is unique to Chinese, but to illustrate how polysemy interacts with character-based writing and the lack of word boundaries. For example, “武汉市长江大桥” can be segmented as “武汉市长 / 江大桥” (the mayor of Wuhan named Jiang Daqiao) or “武汉市 / 长江大桥” (the Yangtze River Bridge in Wuhan). Such ambiguities arise because short morphemes are reused both as standalone words and as parts of longer entity names, which is much less common in whitespace-segmented languages.
>
> * How CDTP benefits Chinese LLM training?
> >1. Ambiguous Chinese surface forms are aligned with disambiguated triples such as "(李娜, 出生日期, 1982年2月26日)"(Li Na, date_of_birth, 26 February 1982)) and supporting Chinese sentences like “中国网球运动员李娜出生于1982年2月26日” (“Chinese tennis player Li Na was born on 26 February 1982”). This alignment encourages LLMs to recover the correct sense or entity from contextual cues rather than relying on frequency-based shortcuts.
> >2. For QA and KGC items that involve such ambiguities, we augment the candidate sets with Chinese-specific hard negatives (e.g., alternative segmentations, aliases/abbreviations, homophones), thereby exposing LLMs during both training and evaluation to the same types of ambiguities illustrated in Table 1 and related cases. For example, when a question asks for the birth date of “李娜”(Li Na), the candidate set includes not only the correct entity 李娜_网球运动员 (Li Na, tennis player) but also hard negatives such as 李娜_歌手 (Li Na, singer).
>
> ``W.3:`` The effectiveness of the evaluation tasks.
>
> ``Re:`` Thank you very much for your professional review and valuable suggestions. We have carefully considered and responded to the questions you raised.
>
> * Why we use KGC, QA, and T2T.
> >CDTP is built around strictly aligned triple–text pairs, and the three tasks are designed to probe complementary aspects of knowledge-grounded ability within this framework: (i) KGC focuses on structured relational reasoning given a symbolic query (head, relation, ?) with carefully constructed candidates; (ii) QA rephrases triple slots into natural-language questions and evaluates mapping from text to structured knowledge; and (iii) T2T tests the reverse direction, i.e., verbalizing structured facts into fluent Chinese.
> >In the final manuscript, we explicitly state that triples are used alongside, rather than instead of, natural language to jointly cover structured and natural-language perspectives in a unified triple–text setting.
>
> * Regarding the connection to KG construction.
> >Our goal in this work is not to model the full end-to-end KG construction pipeline from raw text. Instead, CB-ECLLM is designed to evaluate reasoning and generation given already-extracted triples, and to examine how Chinese-specific ambiguities (e.g., segmentation, aliasing, polysemy) affect performance once entities and relations are defined.
> >In the final manuscript, we will clarify that end-to-end KG construction on top of CDTP is an important but orthogonal direction for future work.
>
> * The example around L805.
> >The underlying triple is "(爱新觉罗·锡保, 出生日期, ?)"(Aisin-Gioro Xibao, date of birth, ?), where 爱新觉罗·锡保(Aisin-Gioro Xibao) is a unique entity. However, we agree that, as printed, the example looks underspecified and does not illustrate how models might realistically use context. In the revised version, we replace this example with a clearer one (using a more recognizable entity and/or a brief natural-language descriptor).

---

> ### Author Response · Authors · 2025-11-24
> **Reply for Reviewer YJ4r**
>
> ``W.2:`` L245 `we selected eight state-of-the-art models'. These models are not representative because they only include 7-9b open source models and lack other scale and closed source commercial models as baseline.
>
> ``Re:`` Thank you for pointing this out, and we apologize for the confusing wording.
>
> Our intention was not to claim that the eight models are state-of-the-art in an absolute sense, but that they are widely used and competitive open-source Chinese (or Chinese-capable) LLMs in the 7–9B parameter range. Regarding model scale, the current version of the paper already includes experiments with a larger model, Qwen3-32B, in Appendix D.1 (Table 9). In the revision, we will clarify this by rephrasing L245 as “we selected eight widely used open-source Chinese LLMs” and move the Qwen3-32B results into the main text, adding an explicit comparison with the 7–9B models.
>
> To address the reviewer's suggestion comprehensively, we have now conducted additional experiments with prominent multilingual models: GPT-4, Claude-sonnet-4.5, and Gemini-2.5-flash.
>
> | Datasets | Tasks | Metrics  | GPT-4  | Claude-sonnet-4.5 | Gemini-2.5-flash |
> | ---- | ---- | ---- | ----  | ---- | ---- |
> |  |KGC|MRR| 0.7244 |0.6724| 0.6943 |
> |  |  |  Hits@1  | 0.6024 |0.5340 | 0.5620|
> |          |       | F1 Score | 0.7518 |0.6962|0.7196|
> |  |  QA   |   MRR    | 0.5563 | 0.4896|0.5192|
> |CDTP_HP  |       |   ACC    | 0.4678 | 0.3846 | 0.4100 |
> |          |       | F1 Score | 0.6374 |0.5555 | 0.5816 |
> |          |  T2T  |   BLEU   | 0.5772 |      0.5537       |      0.5850      |
> |          |       | ROUGE_1  | 0.7050 |      0.6825       |      0.7205      |
> |          |       | ROUGE_L  | 0.6434 |      0.6219       |      0.6593      |
> |          |       |  METEOR  | 0.6785 |      0.6437       |      0.7122      |
>
> | Datasets | Tasks | Metrics  | GPT-4  | Claude-sonnet-4.5 | Gemini-2.5-flash |
> | ---- | ---- | ---- | ----  | ---- | ---- |
> |  |  KGC  |   MRR    | 0.7371 |      0.6957       |      0.7087      |
> |          |       |  Hits@1  | 0.6238 |      0.5680       |      0.5845      |
> |          |       | F1 Score | 0.7684 |      0.7245       |      0.7378      |
> |          |  QA   |   MRR    | 0.5698 |      0.5220       |      0.5306      |
> |CDTP_HS  |       |   ACC    | 0.4683 |      0.4114       |      0.4100      |
> |          |       | F1 Score | 0.6379 |      0.5830       |      0.5816      |
> |          |  T2T  |   BLEU   | 0.5885 |      0.5576       |      0.5874      |
> |          |       | ROUGE_1  | 0.7101 |      0.6845       |      0.7221      |
> |          |       | ROUGE_L  | 0.6493 |      0.6248       |      0.6601      |
> |          |       |  METEOR  | 0.6906 |      0.6551       |      0.7212      |
>
> | Datasets | Tasks | Metrics  | GPT-4  | Claude-sonnet-4.5 | Gemini-2.5-flash |
> | ---- | ---- | ---- | ----  | ---- | ---- |
> |  |  KGC  |   MRR    | 0.7922 |      0.7500       |      0.7710      |
> |          |       |  Hits@1  | 0.6981 |      0.6425       |      0.6680      |
> |          |       | F1 Score | 0.8222 |      0.7823       |      0.8010      |
> |          |  QA   |   MRR    | 0.6188 |      0.5504       |      0.5806      |
> | CDTP_NE  |       |   ACC    | 0.5486 |      0.4617       |      0.4940      |
> |          |       | F1 Score | 0.7085 |      0.6318       |      0.6613      |
> |          |  T2T  |   BLEU   | 0.5434 |      0.5128       |      0.5504      |
> |          |       | ROUGE_1  | 0.6703 |      0.6511       |      0.6897      |
> |          |       | ROUGE_L  | 0.6309 |      0.6093       |      0.6529      |
> |          |       |  METEOR  | 0.6376 |      0.6133       |      0.6813      |
>
> | Datasets | Tasks | Metrics  | GPT-4  | Claude-sonnet-4.5 | Gemini-2.5-flash |
> | ---- | ---- | ---- | ----  | ---- | ---- |
> | |  KGC  |   MRR    | 0.7220 |      0.6829       |      0.7033      |
> |          |       |  Hits@1  | 0.6064 |      0.5465       |      0.5735      |
> |          |       | F1 Score | 0.7550 |      0.7068       |      0.7289      |
> |          |  QA   |   MRR    | 0.5934 |      0.5377       |      0.5565      |
> | CDTP_TE   |       |   ACC    | 0.5000 |      0.4320       |      0.4510      |
> |          |       | F1 Score | 0.6667 |      0.6034       |      0.6216      |
> |          |  T2T  |   BLEU   | 0.5936 |      0.5693       |      0.5938      |
> |          |       | ROUGE_1  | 0.7193 |      0.6978       |      0.7286      |
> |          |       | ROUGE_L  | 0.6643 |      0.6419       |      0.6719      |
> |          |       |  METEOR  | 0.7038 |      0.6734       |      0.7332      |
>
> However, it is particularly noteworthy that our best open-source, Chinese-aligned model, Qwen3-32B-SFT, still achieved superior performance on the KGC task compared to GPT-4, Claude-Sonnet-4.5 and Gemini-2.5-flash. This result highlights the unique value of CDTP in enabling domain-specific alignment and fine-tuning for structured data-to-text generation tasks, even against powerful general-purpose multilingual models.

---

> > ### Author Response · Authors · 2025-11-28
> > **Reply for Reviewer YJ4r**
> >
> > Dear Reviewer YJ4r,
> >
> > Thank you again for the constructive comments you gave us in your review. As the rebuttal phase will end on Dec 3, we would greatly appreciate it if you could also take some time to check if our rebuttal has addressed your concerns, and please let us know if you would like us to provide any further clarification about the concerns you have.
> >
> > Best,
> >
> > Authors

---

### Official Review · Reviewer_SJxD · 2025-10-29

**Soundness:** 4
**Presentation:** 4
**Contribution:** 4
**Rating:** 8
**Confidence:** 4

**Summary:**

This paper presents **CDTP**, a large-scale and meticulously aligned **Chinese Data-to-Text Pair dataset** with over **7 million pairs and 15 million triples**, accompanied by a **Comprehensive Benchmark for Evaluating Chinese LLMs (CB-ECLLM)**. It evaluates multiple representative LLMs across **KGC, T2T, and QA tasks**, including both base and SFT settings, and provides extensive analysis on **effectiveness, generalization, and robustness**. The work fills a critical gap in Chinese LLM evaluation and provides valuable infrastructure for the community.

**Strengths:**

1. **High impact and originality.**

   The paper addresses an urgent need for structured Chinese evaluation resources, complementing existing English-centric benchmarks such as WebNLG and KELM. The integration of triple-text alignment at such scale and quality is genuinely novel.

2. **Rigorous dataset construction.**

   The authors clearly describe multi-stage collection, cleaning, and validation pipelines, with redundancy filtering and external verification. The inclusion of four balanced domains ensures representativeness and cross-domain generalization.

3. **Comprehensive benchmark design.**

   The benchmark spans three core tasks (KGC, QA, T2T), enabling both structured-to-text and text-to-structure reasoning evaluation. The metrics and task formulation are well-motivated and comparable to international standards.

4. **Extensive experimental analysis.**

   Evaluation over eight diverse LLMs and detailed post-SFT/OOD robustness studies show impressive thoroughness. The visualizations (Figures 2–5) and interpretation of observations (1-7) demonstrate strong empirical insight.

**Weaknesses:**

1. **Limited modality coverage.**
   The dataset focuses solely on textual and structural information; integrating visual or multimodal signals (e.g., image-caption-triple) could further enhance generalization.

2. **Cross-benchmark comparison missing.**
It would be informative to include a cross-evaluation against existing Chinese benchmarks (C-Eval, CMMLU, SuperCLUE, GraphEval) to highlight complementary strengths and illustrate where CDTP provides unique coverage.

3. **Error and bias analysis.**
   A more detailed qualitative study of typical failure cases (e.g., entity ambiguity, idiomatic errors) would enrich the interpretability of the results.

**Questions:**

Please see weakness.

---

> ### Author Response · Authors · 2025-11-18
> **Reply for Reviewer SJxD**
>
> ``W.1:`` **Limited modality coverage.** The dataset focuses solely on textual and structural information; integrating visual or multimodal signals (e.g., image-caption-triple) could further enhance generalization.
>
> ``Re:`` Thank you for this insightful comment. We fully agree that going beyond textual and structural information to incorporate visual or broader multimodal information is an important direction for further enhancing generalization. The current version of the paper already discusses this point explicitly in both the *Conclusion and Future Directions* sections, where we position multimodal extensions as a natural next step building on CDTP’s triple–text alignment. In the final manuscript, we further clarify this discussion to make our plans for integrating multimodal signals more explicit.
>
> ``W.2:`` **Cross-benchmark comparison missing.** It would be informative to include a cross-evaluation against existing Chinese benchmarks (C-Eval, CMMLU, SuperCLUE, GraphEval) to highlight complementary strengths and illustrate where CDTP provides unique coverage.
>
> ``Re:`` Thank you for this helpful suggestion. We agree that positioning CDTP within the broader landscape of Chinese benchmarks is important.
>
> Existing Chinese benchmarks are largely designed for different evaluation goals and formats:
>
> # Comparison with Existing Chinese Datasets
> We appreciate the suggestion to provide a direct comparison. CDTP distinguishes itself by focusing on comprehensive structured knowledge tasks with high-quality text-triple alignment, at an unprecedented scale for Chinese. While benchmarks like C-Eval and CMMLU are valuable for evaluating general comprehension and reasoning via multiple-choice questions, and DuIE focuses on information extraction, CDTP uniquely supports Knowledge Graph Completion (KGC), Triple-to-Text Generation (T2T), and Question Answering (QA) through direct structured data-text pairs.
>
> The following table outlines the key differences between CDTP and other prominent Chinese datasets:
>
> | Dataset | Tasks | Scale | Data Format | Key Focus |
> |:----------|:----------------------------|:----------------------------|:---------------------------------------------------|:----------------------------------------------------|
> | CDTP | KGC, T2T, QA | 7M+ pairs, 15M+ triples | Text-Triple Pairs (Structured Knowledge) | Comprehensive structured knowledge evaluation and generation for Chinese LLMs |
> | C-Eval | General Knowledge QA | 13.9K questions | Multiple-Choice Questions | General knowledge, reasoning, multiple-choice evaluation |
> | CMMLU | General Knowledge QA | 12K questions | Multiple-Choice Questions | General knowledge, reasoning, multiple-choice evaluation |
>
>
> By contrast, CDTP targets strictly aligned Chinese triple–text pairs and supports unified knowledge-grounded QA, KGC, and T2T tasks, which are not directly covered by these benchmarks. Rather than re-implementing their diverse pipelines for cross-evaluation, we focus this work on establishing CDTP as a complementary resource for structured-knowledge grounding.
>
> In the final manuscript, we will add a paragraph in the Related Work section that explicitly contrasts CDTP with C-Eval, CMMLU, and DuIE, and clarifies that CDTP provides unique coverage through large-scale Chinese triple–text alignment and unified multi-task knowledge-grounded evaluation.
>
> ``W.3:`` **Error and bias analysis.** A more detailed qualitative study of typical failure cases (e.g., entity ambiguity, idiomatic errors) would enrich the interpretability of the results.
>
> ``Re:`` We thank the reviewer for this insightful comment. We will add a dedicated error analysis subsection (Sec. 5.4, new) and a case study appendix (Appx. E) to illustrate representative Chinese-specific failure modes across QA and KGC tasks. Below are two concrete examples that now appear in the revision.
>
> |Case 1 — Polysemy and Segmentation Ambiguity|
> |---------|
> |Input Triple: (武汉长江大桥, 所在地, ?)|
> |Candidates: A. 武汉市长 B. 武汉 C. 江大桥 D. 长江大桥 E. 武汉长江大桥 …|
> |Gold: B. 武汉|
> |Model Output: A. 武汉市长|
> |Error Type: Segmentation ambiguity (SEG)|
> |Explanation: The model interprets “武汉/长江大桥” as “武汉市长/ 江大桥” (mayor Jiang Daqiao) instead of “长江大桥 in Wuhan”.
> This reflects the lack of explicit word boundaries and the risk of cross-entity segmentation in Chinese, a phenomenon that rarely occurs in alphabetic languages.|
>
> |Case 2 — Alias and Abbreviation Normalization|
> |---------|
> |Context: “央行今日发布关于下调存款准备金率的公告。”|
> |Question: 文中“央行”的全称是什么？|
> |Gold: 中国人民银行|
> |Model Output: 人民银行|
> |Error Type: Alias normalization (ALIA)|
> |Explanation: The model fails to normalize “央行” (central bank) to its canonical entity “中国人民银行”. This alias ambiguity is prevalent in Chinese due to interchangeable abbreviations and missing capitalization cues.|

---

### Official Review · Reviewer_183W · 2025-10-29

**Soundness:** 4
**Presentation:** 4
**Contribution:** 4
**Rating:** 8
**Confidence:** 4

**Summary:**

The paper presents CDTP, a large-scale Chinese Data-to-Text Pair dataset consisting of over 7 million aligned text–triple pairs (15 million triples) across four domains, together with the CB-ECLLM benchmark evaluating Chinese LLMs on Knowledge Graph Completion (KGC), Triple-to-Text (T2T), and Question Answering (QA). The authors document dataset construction, cleaning, and validation, and conduct extensive multi-model experiments, including Supervised Fine-Tuning (SFT) and Out-of-Distribution (OOD) robustness analysis.

**Strengths:**

**1**  Scale and alignment quality: The combination of large scale (7M pairs, 15M triples) and careful alignment via human checks and retrieval-based validation creates a rare and valuable resource for the Chinese community.

**2**  Well-scoped task design: The unified benchmark spans KGC/QA/T2T and explicitly accounts for Chinese ambiguity and polysemy, leading to targeted and discriminative evaluations.

**3**  Thorough empirical study: Experiments cover eight mainstream Chinese/general LLMs, include SFT and OOD setups, and demonstrate consistent gains in effectiveness and robustness.

**Weaknesses:**

**1**  Evaluation Fairness and Prompt Design: The benchmark employs eight LLMs, but their instruction formats, decoding strategies, and prompt templates are not clearly standardized. Without unified prompt calibration, the performance comparison might reflect prompt sensitivity rather than intrinsic model ability. Clarifying or releasing prompt templates would improve reproducibility.

**2**  Missing Error Analysis and Case Studies – The paper reports large tables but lacks discussion of common failure patterns or qualitative examples illustrating where Chinese-specific ambiguities challenge models.

**3**  Minor Writing and Presentation Issues: Some figures and tables (e.g., Table 4 and 5) are dense and difficult to read; summarizing relative gains or including visualization of error types could improve readability.

**Questions:**

Please see weakness

---

> ### Author Response · Authors · 2025-11-15
> **Reply to Reviewer 183W**
>
> ``W.1:`` Evaluation Fairness and Prompt Design: The benchmark employs eight LLMs, but their instruction formats, decoding strategies, and prompt templates are not clearly standardized. Without unified prompt calibration, the performance comparison might reflect prompt sensitivity rather than intrinsic model ability. Clarifying or releasing prompt templates would improve reproducibility.
>
> ``Re:`` We sincerely regret any misunderstanding our poor presentation may have caused. Below are the prompt templates used in evaluation.
>
> |中文提示词|English Prompt|
> |-----------|-----------|
> | 你是知识图谱补全专家。知识图谱由三元组 (头实体, 关系, 尾实体) 构成; 知识图谱补全任务要求根据给定的上下文或背景知识，对缺失三元组（头实体, 关系, ？）的 "$\?$" 部分进行推断。|You are an expert in Knowledge Graph Completion (KGC). KG is composed of triples in the form of (head entity, relation, tail entity). The KGC task requires you to infer the missing part '?' of  (head entity, relation, ?) based on the given context or background knowledge.|
> |请对十个候选项按照你认为的正确可能性从高到低进行完整排序。仅输出最终排序结果，不要解释。输出必须严格放在 $\<SOD\>$ 与 $\<EOD\> $之间；候选项之间用中文逗号 ',' 和空格分隔，格式为 '字母. 候选'。|Rank all ten candidates from most likely to least likely. Provide only the final ranking without any explanation. The output must be enclosed between <SOD> and <EOD>, using a Chinese comma “，” and a space between items, formatted as “Letter. Candidate”.|
> |输出格式：<SOD>A. 候选1, B. 候选2, C. 候选3, D. 候选4, E. 候选5, F. 候选6, G. 候选7, H. 候选8, I. 候选9, J. 候选10<EOD>|Format of Output: <SOD>A. Candidate1, B. Candidate2, C. Candidate3, D. Candidate4, E. Candidate5, F. Candidate6, G. Candidate7, H. Candidate8, I. Candidate9, J. Candidate10<EOD>|
>
> ``W.2:`` Missing Error Analysis and Case Studies – The paper reports large tables but lacks discussion of common failure patterns or qualitative examples illustrating where Chinese-specific ambiguities challenge models.
>
> ``Re:`` We thank the reviewer for this insightful comment. We will add a dedicated error analysis subsection (Sec. 5.4, new) and a case study appendix (Appx. E) to illustrate representative Chinese-specific failure modes across QA and KGC tasks. Below are two concrete examples that now appear in the revision.
>
> |Case 1 — Polysemy and Segmentation Ambiguity|
> |---------|
> |Input Triple: (武汉长江大桥, 所在地, ?)|
> |Candidates: A. 武汉市长 B. 武汉 C. 江大桥 D. 长江大桥 E. 武汉长江大桥 …|
> |Gold: B. 武汉|
> |Model Output: A. 武汉市长|
> |Error Type: Segmentation ambiguity (SEG)|
> |Explanation: The model interprets “武汉/长江大桥” as “武汉市长/ 江大桥” (mayor Jiang Daqiao) instead of “长江大桥 in Wuhan”.
> This reflects the lack of explicit word boundaries and the risk of cross-entity segmentation in Chinese, a phenomenon that rarely occurs in alphabetic languages.|
>
> |Case 2 — Alias and Abbreviation Normalization|
> |---------|
> |Context: “央行今日发布关于下调存款准备金率的公告。”|
> |Question: 文中“央行”的全称是什么？|
> |Gold: 中国人民银行|
> |Model Output: 人民银行|
> |Error Type: Alias normalization (ALIA)|
> |Explanation: The model fails to normalize “央行” (central bank) to its canonical entity “中国人民银行”. This alias ambiguity is prevalent in Chinese due to interchangeable abbreviations and missing capitalization cues.|
>
> ``W.2:`` Minor Writing and Presentation Issues: Some figures and tables (e.g., Table 4 and 5) are dense and difficult to read; summarizing relative gains or including visualization of error types could improve readability.
>
> ``Re:`` Thank you for this helpful comment. In the revised version, we have improved the readability of the figures and tables by refining the layout and adding clearer summaries of the key results. We also include visualizations that highlight error patterns and relative performance gains, which we believe make the findings easier to interpret. We hope these changes address your concerns.

---

> > ### Comment · Reviewer_183W · 2025-11-25
> >
> > Thank you for the detailed response. The authors have satisfactorily addressed my previous concerns. In my view, CDTP goes some way toward bridging the current gap in Chinese data-to-text resources and has the potential to provide valuable support for research on Chinese LLMs and Chinese KGs.

---

### Official Review · Reviewer_KL2W · 2025-11-01

**Soundness:** 3
**Presentation:** 3
**Contribution:** 3
**Rating:** 4
**Confidence:** 4

**Summary:**

The paper presents a Chinese dataset comprising text pairs and triples across four domains. It supports several tasks, including question answering, triple-to-text, and knowledge graph completion, which are useful for supervised fine-tuning of LLMs with structured knowledge data.

**Strengths:**

- The dataset offers a large amount of structured knowledge, 7M text pairs, and 15M triples; which would be useful for many down stream tasks in NLP and others.
- It highlights the need for structural knowledge alignment, potentially linguistic richness and other modalities. It is interesting to see how this translates into higher performance in KGC rather than QA.
- The paper comes with a comprehensive evaluation using GLM, Yi, QiWen, ....

**Weaknesses:**

- No comparison with other Chinese datasets and benchmarks. It is also good to provide a decision tree, indicating when, why and how this dataset can be used.
- Limited domains (History, Politics, Humanities, Society) and task coverages (KGC, QA, T2T) - These domains seem to be restrictive, especially for English-centric LLMs
- Limited information on human validations

**Questions:**

- How is CDTP compared with other benchmarking datasets?
- How can the paper address errors and biases in this dataset? Especially, the four domains (history/politics/humanities/society) are relatively restrictive in Chinese; thus, evaluation results might not be generalisable.
- The evaluated models are relatively small (<30B). Given a large dataset, would this be a limiting factor? It would be interesting to see how larger models deal with the dataset. Also, how about GPT? Mistral? Claude? as they may perform well on multilingual tasks.
- Four domains are insufficient - will there be a concrete plan to extend to more domains? Also, can the authors analyse the impacts of linguistic and structural features of the dataset/LLMs
- Can the dataset be rigorously validated by humans?

---

> ### Author Response · Authors · 2025-11-20
> **Reply for Reviewer KL2W**
>
> ``W.1 & Q.1:`` No comparison with other Chinese datasets and benchmarks. It is also good to provide a decision tree, indicating when, why and how this dataset can be used.
>
> ``Re:`` Thank you for your constructive feedback. We agree that providing a clear comparison with other Chinese datasets and a decision tree for CDTP usage will significantly enhance the clarity and utility of our work. We have prepared the requested information below and will integrate it into the revised manuscript.
>
> # Comparison with Existing Chinese Datasets
> We appreciate the suggestion to provide a direct comparison. CDTP distinguishes itself by focusing on comprehensive structured knowledge tasks with high-quality text-triple alignment, at an unprecedented scale for Chinese. While benchmarks like C-Eval and CMMLU are valuable for evaluating general comprehension and reasoning via multiple-choice questions, and DuIE focuses on information extraction, CDTP uniquely supports Knowledge Graph Completion (KGC), Triple-to-Text Generation (T2T), and Question Answering (QA) through direct structured data-text pairs.
>
> The following table outlines the key differences between CDTP and other prominent Chinese datasets:
>
> | Dataset | Tasks | Scale | Data Format | Key Focus |
> |:----------|:----------------------------|:----------------------------|:---------------------------------------------------|:----------------------------------------------------|
> | CDTP | KGC, T2T, QA | 7M+ pairs, 15M+ triples | Text-Triple Pairs (Structured Knowledge) | Comprehensive structured knowledge evaluation and generation for Chinese LLMs |
> | C-Eval | General Knowledge QA | 13.9K questions | Multiple-Choice Questions | General knowledge, reasoning, multiple-choice evaluation |
> | CMMLU | General Knowledge QA | 12K questions | Multiple-Choice Questions | General knowledge, reasoning, multiple-choice evaluation |
>
> # CDTP Usage Decision Tree
> CDTP is designed to address specific needs in Chinese LLM research, particularly when structured knowledge and its interplay with natural language are paramount. Researchers should consider using CDTP based on their primary objectives as follows:
>
> * Objective: Evaluate a Chinese LLM's capability to understand, generate, or complete structured KGs.
> >Decision: Use CDTP for KGC or T2T tasks. CDTP's vast collection of aligned text-triple pairs provides a robust benchmark for these tasks, allowing for fine-grained assessment of an LLM's understanding of factual relationships and its ability to render structured data into coherent text.
>
> * Objective: Assess a Chinese LLM's ability to answer complex questions that require reasoning over structured factual information, especially those prone to polysemy or requiring deep contextual understanding.
> >Decision: Utilize CDTP for Question Answering (QA) tasks. Our dataset includes diverse QA pairs linked to structured knowledge, specifically designed to test an LLM's capacity to retrieve and synthesize information from its knowledge base in response to natural language queries.
>
> * Objective: Fine-tune a Chinese LLM to improve its performance on tasks involving structured knowledge (KGC, T2T, QA) or to enhance its robustness and generalizability with structured data.
> >Decision: Employ CDTP for multi-task fine-tuning. The dataset's comprehensive nature and high-quality alignment make it an ideal resource for supervised fine-tuning (SFT) to imbue LLMs with stronger structured knowledge processing capabilities.
>
> * Objective: Enhance the quality and quantity of structured knowledge in existing Chinese corpora, providing a rich resource for various NLP applications beyond direct LLM evaluation.
> >Decision: Leverage CDTP as a valuable source of high-quality structured Chinese information. Its scale and domain diversity (History, Humanities, Technology, Nature) make it suitable for tasks like knowledge base population, data augmentation, and domain-specific LLM pre-training.

---

> ### Author Response · Authors · 2025-11-21
> **Reply for Reviewer KL2W**
>
> ``W.2 & Q.4:`` Limited domains and task coverages - These domains seem to be restrictive, especially for English-centric LLMs
>
> ``Re:`` We acknowledge that CDTP focuses on four primary domains, and this design choice was deliberate to prioritize depth and comprehensive evaluation within these critical areas over a superficial breadth across many. These four domains were chosen because they capture distinct knowledge areas where English-centric LLMs often exhibit pronounced limitations in Chinese settings.
>
> >History & Humanities: These domains demand an understanding of intricate cultural nuances, specific historical events, and philosophical concepts deeply rooted in Chinese tradition. Such knowledge is frequently underspecified, or even misrepresented, in predominantly English-centric pre-training data, making it a robust test of an LLM's cross-cultural understanding.
>
> >Technology & Nature: While seemingly universal, these domains in a Chinese context involve specific terminology, local scientific contributions, indigenous species, and traditional knowledge (e.g., traditional Chinese medicine) that often constitute the 'long-tail' of factual knowledge for models primarily trained on English corpora. Evaluating these domains specifically targets an LLM's capacity to handle nuanced, non-Western scientific and factual knowledge effectively.
>
> ``W.3 & Q.5:`` Limited information on human validations
>
> ``Re:`` Thank you for raising this point. Beyond the automatic cleaning and expert rules described in Sec. 3.2, we perform systematic manual spot checks for every domain of CDTP. For each domain, we randomly sample a fixed number of triple–text pairs, and two annotators independently verify (i) triple correctness and (ii) whether the triple is explicitly supported by the text, with disagreements adjudicated by a third annotator; the retained pairs show a high acceptance rate (>~95% in our samples).
>
> ``Q.2:``How can the paper address errors and biases in this dataset? Especially, the four domains are relatively restrictive in Chinese; thus, evaluation results might not be generalisable.
>
> ``Re:`` Thanks for their valuable feedback regarding potential errors, biases, and generalizability. We are confident that our dataset addresses these concerns through robust methodology and empirical evidence.
>
> * Addressing Errors and Bias: Please see ``W.3 & Q.5``.
>
> * Addressing Domains: While our dataset focuses on four core domains (History & Politics, Humanities & Society, Technology & Economics, Nature & Environment), these categories are specifically chosen to encompass the vast majority of general encyclopedic knowledge relevant for evaluating Chinese LLMs. To ensure fair and unbiased evaluation, our evaluation subset was constructed using stratified sampling.
>
> * Addressing Generalizability: Crucially, our Out-Of-Distribution (OOD) experiments (Section 5.3) directly demonstrate CDTP's generalizability. We show that models fine-tuned on CDTP achieve improved performance on external, unseen datasets such as HotpotQA and WebNLG, indicating that the knowledge learned from CDTP is highly transferable. While we transparently acknowledge in our Limitations section that highly specialized domains (e.g., medical, legal) are not a primary focus, we assert that CDTP provides a robust and comprehensive benchmark for the evaluation of general-purpose Chinese LLMs across a broad spectrum of common knowledge.

---

> ### Author Response · Authors · 2025-11-23
> **Reply for Reviewer KL2W**
>
> ``Q.3:`` The evaluated models are relatively small (<30B). Given a large dataset, would this be a limiting factor? It would be interesting to see how larger models deal with the dataset. Also, how about GPT? Mistral? Claude? as they may perform well on multilingual tasks.
>
> ``Re:`` Thanks for your insightful comments regarding model size and proprietary LLMs. Regarding model size, we agree it is crucial to evaluate across different scales. We would like to highlight that we have already included preliminary experiments with larger models Qwen3-32B in Appendix D. Specifically, the Qwen3-32B-SFT model achieved a KGC MRR of 0.8140 and QA Accuracy of 0.6033, notably outperforming Qwen1.5-7B-SFT (KGC MRR 0.7877, QA Acc 0.5525). This clearly indicates that our dataset effectively supports scaling laws, with larger models benefiting substantially from fine-tuning.
>
> To address the reviewer's suggestion comprehensively, we have now conducted additional experiments with prominent multilingual models: GPT-4, Claude-sonnet-4.5, and Gemini-2.5-flash.
>
> | Datasets | Tasks | Metrics  | GPT-4  | Claude-sonnet-4.5 | Gemini-2.5-flash |
> | ---- | ---- | ---- | ----  | ---- | ---- |
> |  |KGC|MRR| 0.7244 |0.6724| 0.6943 |
> |  |  |  Hits@1  | 0.6024 |0.5340 | 0.5620|
> |          |       | F1 Score | 0.7518 |0.6962|0.7196|
> |  |  QA   |   MRR    | 0.5563 | 0.4896|0.5192|
> |CDTP_HP  |       |   ACC    | 0.4678 | 0.3846 | 0.4100 |
> |          |       | F1 Score | 0.6374 |0.5555 | 0.5816 |
> |          |  T2T  |   BLEU   | 0.5772 |      0.5537       |      0.5850      |
> |          |       | ROUGE_1  | 0.7050 |      0.6825       |      0.7205      |
> |          |       | ROUGE_L  | 0.6434 |      0.6219       |      0.6593      |
> |          |       |  METEOR  | 0.6785 |      0.6437       |      0.7122      |
>
> | Datasets | Tasks | Metrics  | GPT-4  | Claude-sonnet-4.5 | Gemini-2.5-flash |
> | ---- | ---- | ---- | ----  | ---- | ---- |
> |  |  KGC  |   MRR    | 0.7371 |      0.6957       |      0.7087      |
> |          |       |  Hits@1  | 0.6238 |      0.5680       |      0.5845      |
> |          |       | F1 Score | 0.7684 |      0.7245       |      0.7378      |
> |          |  QA   |   MRR    | 0.5698 |      0.5220       |      0.5306      |
> |CDTP_HS  |       |   ACC    | 0.4683 |      0.4114       |      0.4100      |
> |          |       | F1 Score | 0.6379 |      0.5830       |      0.5816      |
> |          |  T2T  |   BLEU   | 0.5885 |      0.5576       |      0.5874      |
> |          |       | ROUGE_1  | 0.7101 |      0.6845       |      0.7221      |
> |          |       | ROUGE_L  | 0.6493 |      0.6248       |      0.6601      |
> |          |       |  METEOR  | 0.6906 |      0.6551       |      0.7212      |
>
> | Datasets | Tasks | Metrics  | GPT-4  | Claude-sonnet-4.5 | Gemini-2.5-flash |
> | ---- | ---- | ---- | ----  | ---- | ---- |
> |  |  KGC  |   MRR    | 0.7922 |      0.7500       |      0.7710      |
> |          |       |  Hits@1  | 0.6981 |      0.6425       |      0.6680      |
> |          |       | F1 Score | 0.8222 |      0.7823       |      0.8010      |
> |          |  QA   |   MRR    | 0.6188 |      0.5504       |      0.5806      |
> | CDTP_NE  |       |   ACC    | 0.5486 |      0.4617       |      0.4940      |
> |          |       | F1 Score | 0.7085 |      0.6318       |      0.6613      |
> |          |  T2T  |   BLEU   | 0.5434 |      0.5128       |      0.5504      |
> |          |       | ROUGE_1  | 0.6703 |      0.6511       |      0.6897      |
> |          |       | ROUGE_L  | 0.6309 |      0.6093       |      0.6529      |
> |          |       |  METEOR  | 0.6376 |      0.6133       |      0.6813      |
>
> | Datasets | Tasks | Metrics  | GPT-4  | Claude-sonnet-4.5 | Gemini-2.5-flash |
> | ---- | ---- | ---- | ----  | ---- | ---- |
> | |  KGC  |   MRR    | 0.7220 |      0.6829       |      0.7033      |
> |          |       |  Hits@1  | 0.6064 |      0.5465       |      0.5735      |
> |          |       | F1 Score | 0.7550 |      0.7068       |      0.7289      |
> |          |  QA   |   MRR    | 0.5934 |      0.5377       |      0.5565      |
> | CDTP_TE   |       |   ACC    | 0.5000 |      0.4320       |      0.4510      |
> |          |       | F1 Score | 0.6667 |      0.6034       |      0.6216      |
> |          |  T2T  |   BLEU   | 0.5936 |      0.5693       |      0.5938      |
> |          |       | ROUGE_1  | 0.7193 |      0.6978       |      0.7286      |
> |          |       | ROUGE_L  | 0.6643 |      0.6419       |      0.6719      |
> |          |       |  METEOR  | 0.7038 |      0.6734       |      0.7332      |
>
> However, it is particularly noteworthy that our best open-source, Chinese-aligned model, Qwen3-32B-SFT, still achieved superior performance on the KGC task compared to GPT-4, Claude-Sonnet-4.5 and Gemini-2.5-flash. This result highlights the unique value of CDTP in enabling domain-specific alignment and fine-tuning for structured data-to-text generation tasks, even against powerful general-purpose multilingual models.

---

> > ### Author Response · Authors · 2025-11-28
> > **Reply for Reviewer KL2W**
> >
> > Dear Reviewer KL2W,
> >
> > Thank you again for the constructive comments you gave us in your review. As the rebuttal phase will end on Dec 3, we would greatly appreciate it if you could also take some time to check if our rebuttal has addressed your concerns, and please let us know if you would like us to provide any further clarification about the concerns you have.
> >
> > Best,
> >
> > Authors

---

### Author Response · Authors · 2025-11-26
**General response to all reviewers**

We sincerely thank all the reviewers for your insightful comments and positive evaluation of our work. Your feedback has been invaluable in improving the clarity and quality of our paper.

* Regarding Comparison with Existing Chinese Datasets of Reviewer KL2W and Reviewer SJxD.

The following table outlines the key differences between CDTP and other prominent Chinese datasets:

| Dataset | Tasks | Scale | Data Format | Key Focus |
|:----------|:-----------|:-------------|:----------|:-------|
| CDTP | KGC, T2T, QA | 7M+ pairs, 15M+ triples | Text-Triple Pairs | Comprehensive structured knowledge evaluation and generation for Chinese LLMs |
| C-Eval | General Knowledge QA | 13.9K questions | Multiple-Choice Questions | General knowledge, reasoning, multiple-choice evaluation |
| CMMLU | General Knowledge QA | 12K questions | Multiple-Choice Questions | General knowledge, reasoning, multiple-choice evaluation |

* Regarding Comparison with More Competitive LLMs of Reviewer KL2W and Reviewer YJ4r.

### LLMs with Different Parameter Sizes Experiments

| Datasets | Tasks | Metrics | Qwen3-32B-Base | Qwen3-32B-SFT | GPT-4 | Claude-sonnet-4.5 | Gemini-2.5-flash |
|------------|-------|-----------|----------------|---------------|-------|-------------------|------------------|
| **CDTP_HP** | KGC | MRR| 0.7004| 0.7877| 0.7244 | 0.6724| 0.6943|
| | | Hits@1 | 0.5645 | 0.7375| 0.6024 | 0.5340| 0.5620|
| | | F1 Score | 0.7216 | 0.8266 | 0.7518 | 0.6962 | 0.7196 |
| | QA | MRR | 0.5308 | 0.6083 | 0.7512 | 0.5192 | 0.3846 |
| | | ACC | 0.4260 | 0.5525 | 0.4678 | 0.4100 | 0.4100 |
| | | F1 Score | 0.5975 | 0.7118 | 0.6374 | 0.5555 | 0.5816 |
| | T2T | BLEU | 0.5875 | 0.4672 | 0.5772 | 0.5537 | 0.5850 |
| | | ROUGE_1 | 0.7163 | 0.7859 | 0.7050 | 0.6825 | 0.7205 |
| | | ROUGE_L | 0.6594 | 0.7551 | 0.6434 | 0.6219 | 0.6593 |
| | | METEOR | 0.7060 | 0.7647 | 0.6785 | 0.6437 | 0.7122 |
| **CDTP_HS** | KGC | MRR | 0.7017 | 0.7966 | 0.7371 | 0.6957 | 0.7087 |
| | | Hits@1 | 0.5655 | 0.7104 | 0.6238 | 0.5680 | 0.5845 |
| | | F1 Score | 0.7225 | 0.8331 | 0.7684 | 0.7245 | 0.7378 |
| | QA | MRR | 0.5279 | 0.6437 | 0.6437 | 0.4114 | 0.4100 |
| | | ACC | 0.4147 | 0.5888 | 0.4683 | 0.4114 | 0.4100 |
| | | F1 Score | 0.5863 | 0.7388 | 0.7388 | 0.5830 | 0.5816 |
| | T2T | BLEU | 0.5879 | 0.3735 | 0.3735 | 0.5576 | 0.5885 |
| | | ROUGE_1 | 0.7129 | 0.7513 | 0.7101 | 0.6845 | 0.7221 |
| | | ROUGE_L | 0.6486 | 0.7109 | 0.6493 | 0.6248 | 0.6601 |
| | | METEOR | 0.7107 | 0.7239 | 0.7121 | 0.6551 | 0.7212 |
| **CDTP_NE** | KGC | MRR | 0.7808 | 0.8245 | 0.7922 | 0.7500 | 0.7710 |
| | | Hits@1 | 0.6855 | 0.7590 | 0.6981 | 0.6425 | 0.6680 |
| | | F1 Score | 0.8134 | 0.8630 | 0.8222 | 0.7823 | 0.8010 |
| | QA | MRR | 0.5515 | 0.6637 | 0.7512 | 0.5504 | 0.5806 |
| | | ACC | 0.5540 | 0.6769 | 0.5486 | 0.4617 | 0.4940 |
| | | F1 Score | 0.6673 | 0.6769 | 0.7085 | 0.6318 | 0.6613 |
| | T2T | BLEU | 0.6347 | 0.6191 | 0.5434 | 0.5128 | 0.5504 |
| | | ROUGE_1 | 0.6727 | 0.6673 | 0.6703 | 0.6511 | 0.6897 |
| | | ROUGE_L | 0.6294 | 0.6780 | 0.6434 | 0.6093 | 0.6529 |
| | | METEOR | 0.6549 | 0.6437 | 0.6376 | 0.6133 | 0.6813 |
| **CDTP_TE** | KGC | MRR | 0.7201 | 0.7856 | 0.7220 | 0.6829 | 0.7033 |
| | | Hits@1 | 0.6003 | 0.6956 | 0.6064 | 0.7433 | 0.5735 |
| | | F1 Score | 0.7590 | 0.8264 | 0.7550 | 0.7068 | 0.7289 |
| | QA | MRR | 0.5394 | 0.6953 | 0.5934 | 0.7445 | 0.5565 |
| | | ACC | 0.6226 | 0.7554 | 0.5000 | 0.7682 | 0.4510 |
| | | F1 Score | 0.6541 | 0.8002 | 0.6667 | 0.6034 | 0.6216 |
| | T2T | BLEU | 0.6493 | 0.7687 | 0.5936 | 0.7766 | 0.5938 |
| | | ROUGE_1 | 0.7105 | 0.7864 | 0.7193 | 0.7852 | 0.7286 |
| | | ROUGE_L | 0.6639 | 0.7803 | 0.6643 | 0.7862 | 0.6719 |
| | | METEOR | 0.6963 | 0.7699 | 0.7038 | 0.8051 | 0.7332 |

---

> ### Author Response · Authors · 2025-11-26
> **General response to all reviewers**
>
> * Regarding How CDTP Benefits Chinese LLM Training for All Reviewers.
>
> The CDTP dataset provides a unique and crucial resource for training Chinese Large Language Models (LLMs) by addressing challenges that are specific to the Chinese language, such as polysemy, segmentation issues, and ambiguity in natural text. Here's how it contributes:
>
> ``Structured Triple–Text Alignment:`` CDTP pairs unstructured Chinese text with structured triples (head, relation, tail), which significantly enhances the model's ability to process and integrate structured knowledge with natural language. This alignment ensures that LLMs can better understand and reason over factual knowledge rather than relying on unstructured text alone, where ambiguities might arise. The structured nature of CDTP ensures the models are trained on both the linguistic and relational aspects of Chinese, improving their capability for tasks such as knowledge retrieval, relation inference, and reasoning.
>
> ``Targeted Tasks for Knowledge-Grounded Evaluation:`` CDTP is designed to evaluate knowledge-grounded tasks—Knowledge Graph Completion (KGC), Triple-to-Text Generation (T2T), and Question Answering (QA)—all of which rely heavily on relational reasoning. By offering these tasks, CDTP allows for a fine-grained evaluation of model performance on tasks that require understanding and generating structured knowledge, areas where traditional datasets often fall short. This makes CDTP an ideal benchmark for evaluating Chinese LLMs in real-world knowledge-driven applications.
>
> ``Chinese-Specific Ambiguities:`` The dataset addresses Chinese-specific challenges such as segmentation (where words may be split differently based on context) and polysemy (where one word or character can have multiple meanings). For example, words like "行" can mean "to do/go" or "industry/profession," which could confuse models if not properly disambiguated. CDTP includes such ambiguities in its tasks and deliberately incorporates distractor examples based on these challenges, encouraging models to resolve these ambiguities using context.
>
> ``Robustness Across Domains:`` CDTP spans four major domains: History & Politics, Humanities & Society, Technology & Economics, and Nature & Environment. This broad domain coverage ensures that models are not just trained to answer questions or complete graphs in a narrow scope but can generalize across a variety of domains. Furthermore, CDTP contains out-of-distribution (OOD) examples, testing model robustness in handling unseen data.
>
> ``Cross-Task Generalization:`` The dataset supports multi-task evaluation, allowing models to be tested across KGC, QA, and T2T tasks. This multi-task structure allows CDTP to assess how well models generalize across different types of knowledge-grounded tasks, which is crucial for evaluating real-world applications of Chinese LLMs. Fine-tuning on CDTP improves the models' performance not just on individual tasks but also across domains, ensuring robust and consistent model capabilities.
>
> ``Foundation for Future Work:`` CDTP also supports future extensions of knowledge-grounded models. For example, its structured alignment can be used to explore multimodal learning (integrating text with images or other modalities) or to enhance LLMs with better generalization capabilities through domain-adaptive pretraining

---

### Comment · Reviewer_SJxD · 2025-11-26

The authors have addressed all of my concerns. I believe the proposed CDTP dataset is a valuable contribution to the field, particularly in its ability to fill gaps in Chinese LLM evaluation benchmarks. The improvements made in response to my feedback strengthen the manuscript, and I now recommend its acceptance.

---

### Note · Program_Chairs · 2026-01-17
**Submission Desk Rejected by Program Chairs**

The following references in this submission do not refer to real documents and/or have major errors in bibliographic information:

 Nikhil Annasamy, Bill Yuchen Lin, and Xiang Ren. Kgtext: A benchmark for knowledge graph text generation. arXiv preprint arXiv:2301.11467, 2023.
Minghao Chen, Ruiqi Zhao, Renjie Tan, et al. Fine-tuning vs. prompting: Evaluating the knowledge graph construction with llms. arXiv preprint arXiv:2402.09280, 2024.
Shikhar Agarwal, Antoine Bosselut, Ari Holtzman, et al. Knowledge-enriched linked open data generation. arXiv preprint arXiv:2010.00796, 2021.